# STROKE3D: LIFTING 2D STROKES INTO RIGGED 3D MODEL VIA LATENT DIFFUSION MODELS

**Ruisi Zhao**[1], **Haoren Zheng**[1], **Zongxin Yang**[2], **Hehe Fan**[1], **Yi Yang**[1]*
[1]ReLER, CCAI, Zhejiang University    [2]DBMI, HMS, Harvard University

{zhaors00, zhenghaoren, hehefan, yangyics}@zju.edu.cn

Zongxin_Yang@hms.harvard.edu

https://whalesong-zrs.github.io/Stroke3D_project_page/

## ABSTRACT

Rigged 3D assets are fundamental to 3D deformation and animation. However, existing 3D generation methods face challenges in generating animatable geometry, while rigging techniques lack fine-grained structural control over skeleton creation. To address these limitations, we introduce *Stroke3D*, a novel framework that directly generates rigged meshes from user inputs: 2D drawn strokes and a descriptive text prompt. Our approach pioneers a two-stage pipeline that separates the generation into: 1) **Controllable Skeleton Generation,** where we employ the Skeletal Graph VAE (*Sk-VAE*) to encode the skeleton's graph structure into a latent space, and the Skeletal Graph DiT *(Sk-DiT)* generates a skeletal embedding. The generation process is conditioned on both the text for semantics and the 2D strokes for explicit structural control, with the VAE's decoder reconstructing the final high-quality 3D skeleton; and 2) **Enhanced Mesh Synthesis via TextuRig and SKA-DPO,** where we then synthesize a textured mesh conditioned on the generated skeleton. For this stage, we first enhance an existing skeleton-to-mesh model by augmenting its training data with *TextuRig*—a dataset of textured and rigged meshes with captions, curated from Objaverse-XL. Additionally, we employ a preference optimization strategy, *SKA-DPO*, guided by a skeleton-mesh alignment score, to further improve geometric fidelity. Together, our framework enables a more intuitive workflow for creating ready-to-animate 3D content. To the best of our knowledge, our work is the first to generate rigged 3D meshes conditioned on user-drawn 2D strokes. Extensive experiments demonstrate that Stroke3D produces plausible skeletons and high-quality meshes.

## 1 INTRODUCTION

Rigged 3D assets (Au et al., 2008; Xu et al., 2019) are fundamental to realistic 3D deformation and animation (Zhao et al., 2024; 2025). They are widely applied in AR/VR, robotics simulation, and the film industry. Professional 3D creation software, such as Blender (Blender, 2025), provides powerful tools for precise geometry creation and animation control. However, the steep learning curve and complexity of these platforms often present a significant barrier for beginners. As a result, numerous generative methods have emerged that offer more user-friendly alternatives to these tools.

Although existing methods achieve notable results in 3D asset generation, they still face two key limitations. **1) *Difficulty in generating animatable geometry.*** Numerous studies (Shi et al., 2023; Zhang et al., 2024; Lai et al., 2025) focus on generating 3D representations, but these methods typically yield static geometries that lack the skeletal hierarchy required for animation. One line of work, such as SKDream (Xu et al., 2025), which generates animatable meshes conditioned on skeletons, is limited by the scarcity of datasets with high-quality, paired skeleton and textured-mesh annotations. **2) *Limited structural control over the skeleton creation.*** Current methods for skeleton generation (Song et al., 2025b; Deng et al., 2025; Zhang et al., 2025) typically employ an end-to-end mesh-to-skeleton paradigm, conditioned on large skeleton-mesh datasets. However, the absence

---

*Corresponding author

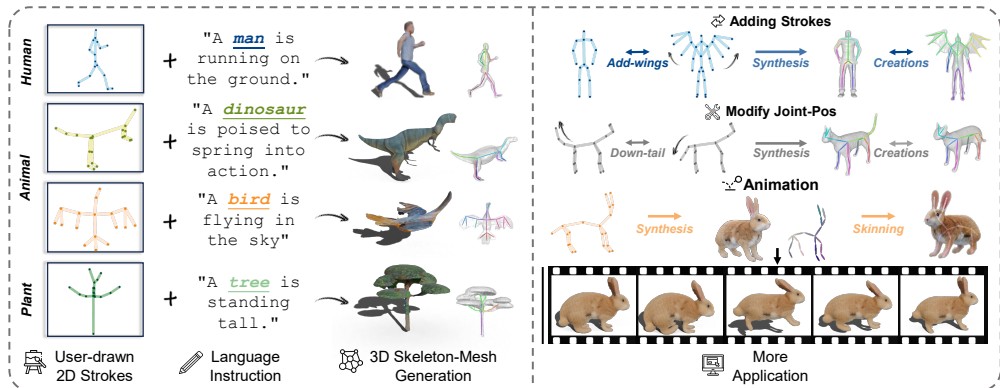

Figure 1: We present **Stroke3D (§Section 3)**, a novel framework that generates rigged 3D meshes from user-drawn strokes and language instructions. We show versatile downstream applications, including generation from different viewpoints, structural editing by adding strokes or modifying joint positions, and final animation. Skeleton color represents the depth in 3D space.

of explicit structural constraints leads to skeletons appearing in unnecessary locations while being absent where they are critically needed, resulting in unpredictable results.

Considering these limitations, we introduce *Stroke3D*, a novel framework that enables users to generate rigged 3D meshes simply by drawing 2D strokes and providing a text prompt. Unlike previous methods that generate the mesh first and then rig it, our method pioneers a skeleton-driven workflow, empowering non-professional users to effortlessly generate ready-to-animate 3D assets. **For controllable skeleton generation**, our method represents the skeleton as a graph and directs *a latent graph diffusion model* (Ho et al., 2020; Song et al., 2020; 2021) to synthesize a complete 3D skeleton conditioned on user input. **To facilitate the mesh generation**, we introduce *TextuRig*, a curated dataset from Objaverse-XL (Deitke et al., 2023). We leverage it to augment training data for skeleton-to-mesh method SKDream (Xu et al., 2025) to create the final mesh. We then apply *SKA-DPO*, a custom DPO (Rafailov et al., 2023) strategy that significantly enhances the final mesh's geometric quality and correspondence to the skeleton.

More specifically, Stroke3D builds upon a Skeletal Graph VAE (Kipf & Welling, 2016; Kingma & Welling, 2013) (*Sk-VAE*) that encodes the skeletal graph structure into latent space. To generate skeleton embeddings, we introduce a Skeletal Graph Diffusion Transformer (Peebles & Xie, 2023) (*Sk-DiT*) that operates within this latent space. We employ TransformerConv (Shi et al., 2020) to perform self-attention (Vaswani et al., 2017), which is suited for graph-structured data. Furthermore, our Sk-DiT incorporates cross-attention to integrate semantic guidance from the text. For structural conditioning, the model is guided by concatenating features derived from the user-drawn strokes with the latent noise. To simulate the stroke input during training, we create stroke-like data by applying perturbations to 2D projections of 3D skeletons, which mimics the imprecision inherent in hand-drawn strokes. Finally, the VAE's decoder reconstructs the 3D skeleton from the generated embedding, ensuring that the output is both realistic and aligned with the user's specified intent.

For the mesh synthesis stage, we present *TextuRig*, a dataset comprising textured, rigged 3D models with detailed captions. Our curation process builds upon the rigged subset of Objaverse-XL (Deitke et al., 2023) identified by UniRig (Zhang et al., 2025). However, observing that the existing data in UniRig often lacks essential texture, we execute a specialized re-processing pipeline. This involved a filtering step to strictly verify the presence of texture maps or vertex colors, followed by re-captioning each model via Gemini (Team et al., 2023) to provide rich textual context. We utilize TextuRig to augment the training data for the SKDream model, thereby enhancing its generative robustness. To further improve geometric fidelity, we apply *SKA-DPO*, a tailored preference optimization strategy. This method leverages the SKA Score (**SK**eleton **A**lignment Score) (Xu et al., 2025) to evaluate skeleton-mesh alignment, constructing preference pairs for Direct Preference Optimization (DPO) (Rafailov et al., 2023) that guide the model toward superior structural coherence.

Combining the designs mentioned above, Stroke3D achieves superior performance in generating high-quality skeleton-mesh pairs. Furthermore, the generations can be directly processed by mature and standard automatic skinning tools (e.g., in Blender) to obtain fully rigged assets. Quantitative evaluations on the MagicArticulate and SKDream benchmarks further demonstrate our superior

Figure 2: **Skeleton-Caption Pipeline (§Section 3.1).** We render the skeleton and its corresponding mesh together into orthogonal projections using pyrender or Blender. This provides the necessary visual context to represent the object's identity and pose clearly. These views are then fed into a Vision-Language Model (VLM) to generate detailed descriptions of the object's identity and pose.

performance on skeleton and mesh generation, respectively. Stroke3D excels in skeleton generation, achieving the lowest Chamfer Distance on most metrics. In particular, our method improves the $\text{Mean}_{Inst.}$ SKA score by **nearly 10 points** over the SKDream baseline. In summary, our main contributions are as follows:

- **To the best of our knowledge, we are the first to generate rigged 3D meshes directly from user-drawn 2D strokes and text prompts.** Our Stroke3D pioneers a skeleton-first generation pipeline that utilizes a *latent graph diffusion model* to synthesize a 3D skeleton.

- For mesh generation, we introduce *TextuRig*, a dataset of textured and rigged 3D meshes with captions, which we leverage to improve the quality of generated textured meshes. Additionally, we apply *SKA-DPO* to further refine the model's geometric fidelity and alignment.

- We demonstrate through extensive experiments that Stroke3D achieves superior performance. Our method consistently outperforms existing approaches in generating plausible skeletons and high-fidelity rigged meshes that align with user inputs.

## 2 RELATED WORK

**3D Generation.** Diffusion models (Ho et al., 2020; Song et al., 2020; 2021) demonstrate remarkable success in vision generation (Rombach et al., 2022; Wan et al., 2025; Labs, 2024; Wu et al., 2024a;b; 2025b). This success also catalyzes significant progress in the field of 3D generation (Li et al., 2023b; Zhou et al., 2025; Li et al., 2026). In early work, DreamFusion (Poole et al., 2022) is the first to use a SDS loss to optimize a NeRF (Mildenhall et al., 2021) representation. Subsequently, many methods adopt a multi-stage approach (Li et al., 2023a; Xu et al., 2024). MVDream (Shi et al., 2023) first generates multiple views, which are then used by a reconstruction model such as LRM (Hong et al., 2023) to create the final 3D asset. More recent works (Xiang et al., 2025; Lai et al., 2025), such as CLAY (Zhang et al., 2024), use a transformer-based 3D diffusion framework (Peebles & Xie, 2023) and various conditions such as images and point clouds to generate 3D representations.

**Skeleton Generation.** Mesh rigging (Au et al., 2008; Xu et al., 2019; Baran & Popović, 2007; Li et al., 2021) represents a long-standing area of exploration. Early works such as RigNet (Xu et al., 2020) predict skeletons directly from the 3D mesh. SKDream (Xu et al., 2025) uses methods based on MCF (Tagliasacchi et al., 2012) to generate curve skeletons from meshes. More recently, a new trend emerges with methods (Song et al., 2025b; Liu et al., 2025; Deng et al., 2025; Zhang et al., 2025) that propose auto-regressive skeleton generation, which is supported by the large-scale skeleton-mesh datasets, enabling more generalizable solutions for the community. However, decoupling skeleton creation from geometry generation is a shared limitation of these methods, resulting in a lack of explicit control over the final rigged structure and often leading to unpredictable outcomes.

**RL in Diffusion.** The remarkable advancements in Large Language Models (LLMs) (Bai et al., 2022a;b; Lee et al., 2023) and Vision-Language Models (VLMs) (Sun et al., 2023; Yu et al., 2024a;b) demonstrate the power of Reinforcement Learning. Inspired by this success, diffusion models (Li et al., 2025a; Ye et al., 2024; Li et al., 2025b; Wu et al., 2025a) leverage RL tools to achieve higher quality generation with human preference. DiffusionDPO (Wallace et al., 2024) optimizes diffusion models using human preference data, and DPOK (Fan et al., 2023) integrates policy optimization with Kullback-Leibler (KL) regularization. In 3D generation, Carve3D (Xie et al., 2024) leverages

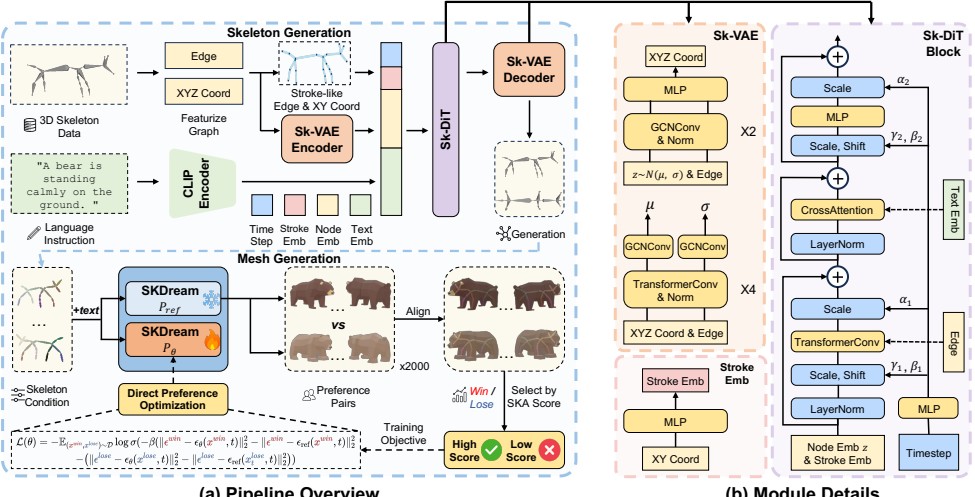

Figure 3: **Overview of Stroke3D (§Section 3).** During the training phase, Sk-VAE encodes a skeleton graph (XYZ coordinates and edges) into a latent space. Subsequently, Sk-DiT is trained to generate these latent embeddings, conditioned on the corresponding 2D strokes and text prompt. After training with TextuRig, we leverage SKA-DPO to further refine SKDream with a skeleton-mesh alignment reward signal. The right side illustrates the implementation details of our models.

multi-view consistency as a reward signal to finetune the model, and DreamDPO (Zhou et al., 2025) provides automated evaluations by VLM for its DPO framework. In the domain of rigging, Auto-Connect (Guo et al., 2025) demonstrates that its stragety can achieve improved rigging quality.

## 3 METHOD

As illustrated in Fig. 3, our proposed framework, Stroke3D, is designed to translate user-drawn 2D strokes and text prompts into rigged 3D meshes. The implementation process begins with **Data Preparation (§Section 3.1)**, where we provide descriptive text labels for each 3D skeleton, curate the TextuRig dataset by sourcing assets from Objaverse-XL (Deitke et al., 2023) and design a canvas tool to record skeleton topology through user-drawn strokes during inference. Subsequently, a **Skeletal Graph Variational Autoencoder (§Section 3.2)** is trained to learn a continuous latent space for these skeletal graphs. Operating within latent space, a **Skeletal Graph Diffusion Transformer (§Section 3.3)** then generates a skeletal embedding that adheres to the user's input. The final stage is **Enhanced Mesh Synthesis via TextuRig and SKA-DPO (§Section 3.4)**. Here, we first leverage the *TextuRig* dataset to augment the SKDream (Xu et al., 2025). We then apply *SKA-DPO*, which uses the skeleton-mesh alignment quality as a reward signal to achieve DPO (Rafailov et al., 2023). This process ensures that the final generated mesh achieves high structural coherence.

### 3.1 DATA PREPARATION

Our data preparation pipeline is composed of three critical stages: 1) enriching existing skeleton datasets with detailed semantic captions, 2) curating a new dataset, which we term TextuRig, to facilitate high-quality skeleton-to-mesh generation, and 3) recording skeleton through designed canvas tool during inference.

**Skeleton Caption.** Existing datasets for skeleton generation (Xu et al., 2025; Song et al., 2025b) often provide only coarse category labels, such as "character" or "animal". Considering this limitation, we develop a pipeline to automatically generate rich, descriptive captions. Directly generating detailed descriptions from skeleton renderings is challenging, as the skeleton's structure are often too abstract for accurate interpretation. As illustrated in Fig. 2, the skeleton of a mushroom can appear as a simple stick-like structure, making its identity unrecognizable. To address this, we first render orthogonal projections of each 3D model using pyrender or Blender (Blender, 2025), displaying both the mesh and its corresponding skeleton together. This approach provides the necessary visual context to make the object's identity and pose clearly discernible. We then leverage a powerful Vi-

sion Language Model (VLM), such as GPT-4 (Achiam et al., 2023) or Gemini (Team et al., 2023), to automatically generate descriptive captions. We feed the rendered orthogonal views into the VLM with a specific prompt, instructing it to describe the object's identity and pose in detail.

**TextuRig Curation.** To generate high-quality meshes from skeletons, 3D models must have both consistent rigging and texture data. These features are not consistently available in the large-scale Objaverse-XL (Deitke et al., 2023) dataset, which is dominated by static assets. To build a suitable dataset, we develop a targeted curation pipeline. Our process begins with the subset of Objaverse-XL filtered by UniRig (Zhang et al., 2025), which provides a collection of models already equipped with skeletons. However, since their processed data are often deficient in corresponding textures, we re-process the original raw assets to preserve the material information. Building upon this foundation, we introduce an additional filtering step to further isolate models that also have high-quality texture information by inspecting for the presence of mesh vertex colors or a texture map. This ensures that our final dataset, TextuRig, contains assets suitable for training a texture-aware, skeleton-driven mesh generation model. To complete the dataset, we employ a similar VLM-based captioning method described previously to generate detailed descriptions of each model's category and pose, creating a rich, multi-modal resource for our research.

**Canvas Interface and Data Recording.** Our stoke-recording pipeline utilizes a specialized canvas tool, deliberately designed to mirror the standard representation of skeletons in professional 3D software (e.g., Blender (Blender, 2025)). In these 3D environments, a skeleton is fundamentally defined not as a sketch, but as a set of discrete joints (points) connected by rigid bones (line segments). To strictly align with this paradigm, our tool guides users to interact by clicking to instantiate 2D joints and creating connected strokes. This process naturally enforces the creation of a structured graph $\mathcal{G}_{\mathbf{2D}} = (\mathbf{J_{xy}}, \mathbf{E})$, where the user-placed points directly correspond to the joints $\mathbf{J_{xy}}$ and the connecting lines explicitly define the topological edges $\mathbf{E}$ that are shared by both the 2D input and the target 3D skeleton. By ensuring the input format is topologically isomorphic to the target 3D representation, we effectively bridge the domain gap between 2D user guidance and 3D generation.

## 3.2 Skeletal Graph Variational Autoencoder

**Learning Structural Embeddings.** We represent a 3D skeleton as an undirected graph $\mathcal{G} = (\mathbf{X}, \mathbf{E})$, where $\mathbf{X} \in \mathbb{R}^{N \times 3}$ represents the 3D coordinates of the $N$ joints, and $\mathbf{E}$ is the set of edges defining the skeleton's topology. To enhance the structural information in each node, we embed its 3D coordinates into a high-dimensional space, transforming the isolated positional data into a dense representation that captures local context. Our Sk-VAE encoder allows features to be aggregated and updated between neighboring nodes, enabling the latent embedding for each joint to implicitly contain structural information. The architecture is composed of GCN (Kipf, 2016) and Transformer-Conv (Shi et al., 2020) for graph data structure.

**Training Objective.** Given an input skeleton graph $\mathcal{G} = (\mathbf{X}, \mathbf{E})$, the encoder network $\mathcal{E}$ maps the graph to a latent distribution, from which we sample a representation $z = \mathcal{E}(\mathbf{X}, \mathbf{E})$. The decoder network $\mathcal{D}$ then reconstructs the skeleton's joint coordinates from this latent code, conditioned on the graph topology $\mathbf{E}$ (which is derived from the input during training or user-drawn strokes during inference), yielding $\hat{\mathbf{X}} = \mathcal{D}(z, \mathbf{E})$. Inspired by Stable Diffusion (Rombach et al., 2022), we regularize the latent space using a slight KL divergence penalty to avoid arbitrarily high-variance latent spaces. This term encourages the learned posterior distribution to approximate a standard Gaussian distribution $\mathcal{N}(0, I)$. The model is trained by minimizing a combined objective function, which includes this KL-divergence term and an $L_2$ reconstruction loss between the input $\mathbf{X}$ and the reconstructed output $\hat{\mathbf{X}}$. The complete training objective is provided in the Appendix.

## 3.3 Skeletal Graph Diffusion Transformer

**Model Architecture.** For latent generation, we adopt an architecture that is based on the design of DiT (Peebles & Xie, 2023), inspired by pioneering works (Labs, 2024; Wan et al., 2025) in generative modeling. We replace the standard self-attention layer with TransformerConv (Shi et al., 2020), which is more suitable for graph-structured data as it ensures the attention mechanism operates only between nodes connected by an edge. To achieve semantic guidance, we first encode the text prompt using a CLIP (Radford et al., 2021) encoder. Then, within each layer of our Sk-DiT, every node em-

bedding performs cross-attention (Vaswani et al., 2017) with the resulting text embedding, allowing the generation to be guided by the user's description.

**Structural Guidance via 2D Strokes.** Directly generating 3D skeletons from text presents a significant challenge, as different categories of subjects often possess varying numbers of joints and distinct topologies. To overcome this structural ambiguity, we propose a method that uses user-drawn strokes to provide explicit structural guidance for the generation process. Our framework allows a user to draw a desired skeleton structure using a simple provided canvas tool. As the user draws, the canvas tool records the 2D coordinates of each joint and the connections between them, effectively defining a complete graph topology for the target skeleton. To train a model that can leverage this guidance, we simulate the stroke data by creating 2D projections of the ground-truth 3D skeletons, which are then perturbed to better reflect the nuances of hand-drawn input. During the generation process, we map these 2D coordinates into a feature space and concatenate them with the noisy latent embedding. By conditioning the generation in this manner, the model learns to produce 3D skeletons whose structure and pose closely conform to the shape of the 2D strokes provided. This ensures that the final output aligns precisely with the user's intent.

The generation process can be formulated as learning a denoiser $\epsilon_\phi$ that predicts the noise $\epsilon$ from the noisy latent embedding $\mathbf{z}_t$ at timestep $t$, conditioned on the joint XY-coordinates embeddings $\mathbf{J}_{xy}$, edge topology $\mathbf{E}$, and the text embedding $\mathbf{c}_{\text{text}}$. To effectively leverage the textual condition, we incorporate classifier-free guidance (CFG) (Ho & Salimans, 2022) during training. The final objective function is as follows:

$$\mathcal{L}_{\text{Sk-DiT}} = \mathbb{E}_{\mathbf{z}_0, t, \epsilon, \mathbf{J}_{xy}, \mathbf{E}, \mathbf{c}_{\text{text}}} \left[ \|\epsilon_\phi(\mathbf{z}_t, t, \mathbf{J}_{xy}, \mathbf{E}, \mathbf{c}_{\text{text}}) - \epsilon\|_2^2 \right] \tag{1}$$

### 3.4 Enhanced Mesh Synthesis via TextuRig and SKA-DPO

**Data augmentation with TextuRig.** The original SKDream model (Xu et al., 2025) utilizes Mean Curvature Flow (MCF) (Tagliasacchi et al., 2012) to generate curve skeletons from meshes, which are subsequently used to fine-tune MVDream (Shi et al., 2023) for skeleton-conditioned multi-view generation. However, the efficacy of this method is constrained by low-quality training data. While existing large-scale rigging datasets (Liu et al., 2025; Song et al., 2025b) provide high-quality skeletal structures, they are often deficient in corresponding textures. Considering these limitations, we leverage our curated TextuRig dataset to augment the training data. Specifically, we utilize the rich textual descriptions generated for TextuRig as prompts, injecting them as semantic conditions into the model consistent with the MVDream paradigm. We then apply a Supervised Fine-Tuning (SFT) stage to MVDream following SKDream, thereby achieving superior generation.

**Skeleton-mesh alignment guided optimization.** To further enhance the structural coherence between the generated mesh and its conditioning skeleton, we introduce a preference-guided optimization strategy. This process begins by employing a reference model, $p_{ref}$, to generate a pair of multi-view candidates for a given skeleton $s$ and text prompt $c$. Subsequently, each candidate is evaluated using a fine-tuned Dinov2 (Oquab et al., 2023) model from SKDream (Xu et al., 2025), which yields a SKA Score quantifying the geometric correspondence between the mesh and the skeleton.

Through a comparative analysis of the SKA Scores, we designate the higher-scoring sample as the winner $x^{win}$ and the other as the loser $x^{lose}$, thereby constructing a preference dataset $\mathcal{D} = \{(c, s, x^{win}, x^{lose})\}$. This dataset is then leveraged to fine-tune a new model, $p_\theta$, with the objective of internalizing these structural preferences. By fine-tuning on these explicit preferences, the model $p_\theta$ learns to produce geometries that exhibit superior skeleton-mesh alignment, consequently surpassing the generation quality of the original reference model, $p_{ref}$. The training objective follows DiffusionDPO (Wallace et al., 2024):

$$\mathcal{L}(\theta) = -\mathbb{E}_{(x^{win}, x^{lose}) \sim \mathcal{D}} \log \sigma(-\beta(\|\epsilon^{win} - \epsilon_\theta(x_t^{win}, t)\|_2^2 - \|\epsilon^{win} - \epsilon_{\text{ref}}(x_t^{win}, t)\|_2^2$$
$$- (\|\epsilon^{lose} - \epsilon_\theta(x_t^{lose}, t)\|_2^2 - \|\epsilon^{lose} - \epsilon_{\text{ref}}(x_t^{lose}, t)\|_2^2)) \tag{2}$$

where $t$ is the timestep in the diffusion process, $\beta$ is a weighting parameter, $\epsilon^{win}$ and $\epsilon^{lose}$ denote the Gaussian noises for $x_t^{win}$ and $x_t^{lose}$ respectively. Intuitively, this objective encourages the model $\epsilon_\theta$ to more accurately predict the noise for winning samples ($x_t^{win}$) and less accurately for losing samples ($x_t^{lose}$), relative to the reference model.

Table 1: **Quantitative comparison of Chamfer Distance (CD) (§Section 4.3).** CD scores are calculated over three metrics (CD-J2J, CD-J2B, CD-B2B) across three categories. The lowest and second-lowest scores are shown in bold and underlined, respectively.

| Method\CD Score ↓ | CD-J2J | | | | CD-J2B | | | | CD-B2B | | | |
|---|---|---|---|---|---|---|---|---|---|---|---|---|
| | All | Character | Animal | Plant | All | Character | Animal | Plant | All | Character | Animal | Plant |
| RigNet [SIGGRAPH 20] | 0.078 | 0.061 | 0.092 | 0.089 | 0.066 | 0.047 | 0.080 | 0.075 | 0.065 | 0.046 | 0.081 | 0.069 |
| SKDream [CVPR25 highlight] | 0.111 | 0.091 | 0.098 | 0.157 | 0.092 | 0.073 | 0.074 | 0.122 | 0.083 | 0.068 | 0.064 | 0.102 |
| MagicArti. [CVPR25] | 0.052 | **0.032** | 0.070 | 0.079 | 0.041 | **0.024** | 0.055 | 0.051 | **0.034** | **0.023** | 0.047 | 0.039 |
| UniRig [SIGGRAPH25] | 0.063 | 0.055 | 0.076 | 0.085 | 0.051 | 0.044 | 0.060 | 0.061 | 0.041 | 0.034 | 0.049 | 0.046 |
| **Ours** | **0.048** | 0.039 | **0.053** | **0.040** | **0.039** | 0.031 | **0.043** | **0.027** | 0.034 | 0.028 | **0.036** | **0.021** |

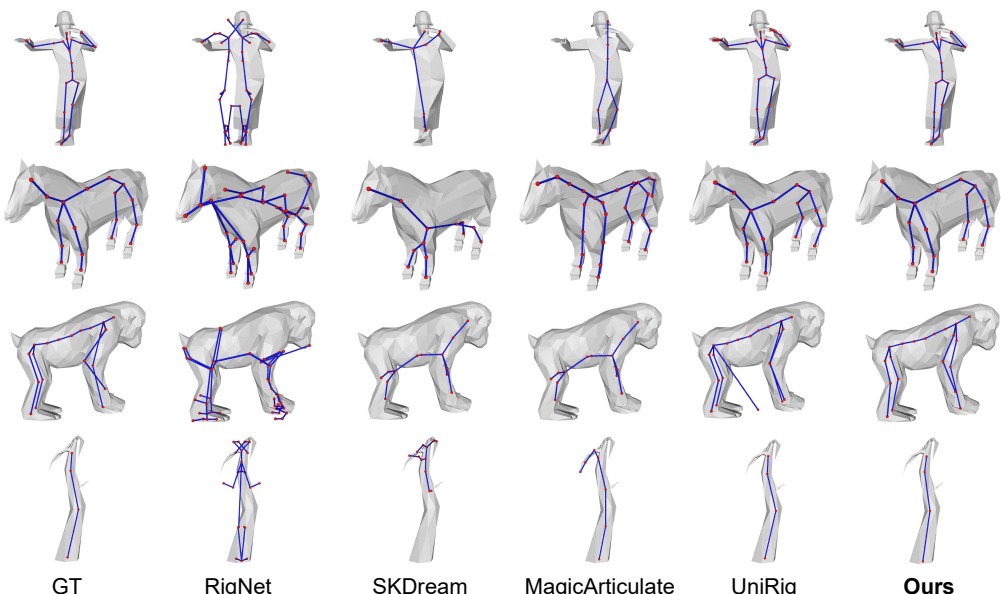

| GT | RigNet | SKDream | MagicArticulate | UniRig | **Ours** |

Figure 4: **Qualitative comparison of skeleton generation (§Section 4.3).** Unlike existing rigging methods that take 3D meshes as input, our approach utilizes 2D projections of a 3D skeleton. This method produces plausible skeletons that more faithfully adhere to the ground truth.

# 4 EXPERIMENT

## 4.1 IMPLEMENT DETAILS

**Training Protocol for Skeleton Generation.** We train our model on the MagicArticulate dataset (Song et al., 2025b). To ensure high-quality structural learning, we curate a representative subset by filtering for models with 0-30 skeletal nodes, which covers the vast majority of effective articulated assets found in real-world applications. Both the Sk-VAE and the Sk-DiT are trained for 500k iterations using the AdamW optimizer with a learning rate of $1 \times 10^{-4}$. For the Sk-VAE, we use a batch size of 64 and a slight KL penalty ($kl\_\beta = 1 \times 10^{-8}$), while the Sk-DiT is trained with a batch size of 128. All experiments are conducted on a NVIDIA A100 GPU (40GB).

**Training Protocol for Mesh Synthesis.** Due to the heterogeneous data sources of the MagicArticulate dataset (Song et al., 2025b) making texture retrieval difficult, we instead utilize the SKDream dataset augmented with our curated TextuRig. While the SKDream dataset contains approximately 24,000 samples, our curated TextuRig augments this with an additional 6,800 high-quality skeleton and texture annotations. Examples from SKDream and TextuRig are shown in Fig. 6. Following the original SKDream training setting, we initialize the diffusion model with pre-trained MVDream (Xu et al., 2025) weights. We then fine-tune the model using our combined dataset, training for 9,000 steps with a learning rate of $1 \times 10^{-5}$. In the SKA-DPO stage, we sample 2,000 skeleton-caption pairs from the training set. For each pair, two multi-view candidates are generated using different random noise seeds. The SKA Score is calculated for each candidate to identify the preferred and dispreferred samples, thereby constructing our preference dataset. The model is fine-tuned on this dataset for 1,000 steps with a learning rate of $5 \times 10^{-6}$.

Table 2: **Quantitative comparison of SKA score (§Section4.3).** Scores are calculated over three classes and three subclasses of animal. The highest scores are bold and the second highest are underlined. The rows *+TextuRig* and *+SKA-DPO* indicate the integration of our curated dataset and preference optimization strategy into the SKDream baseline, respectively.

| Method\SKA Score ↑ | $\text{Mean}_{Inst.}$ | $\text{Mean}_{Class}$ | Character | Animal | Plant | Apodes | Bipeds | Wings |
|---|---|---|---|---|---|---|---|---|
| SDEdit [ICLR22] | 72.11 | 67.32 | 67.60 | 80.91 | 53.46 | 79.73 | 83.17 | 73.82 |
| SKDream [CVPR25 highlight] | 80.43 | 74.38 | 78.45 | 91.16 | 53.53 | 94.47 | 85.20 | 88.40 |
| *+TextuRig* | 82.37 | 76.84 | 84.80 | 91.73 | 53.99 | **94.49** | 86.43 | 92.36 |
| *+SKA-DPO* | 85.57 | 81.12 | 83.56 | 93.35 | 66.75 | 94.48 | **89.75** | 93.64 |
| **Ours** (*+TextuRig & SKA-DPO*) | **87.83** | **84.36** | **88.70** | **93.75** | **70.63** | 93.33 | 89.29 | **94.55** |

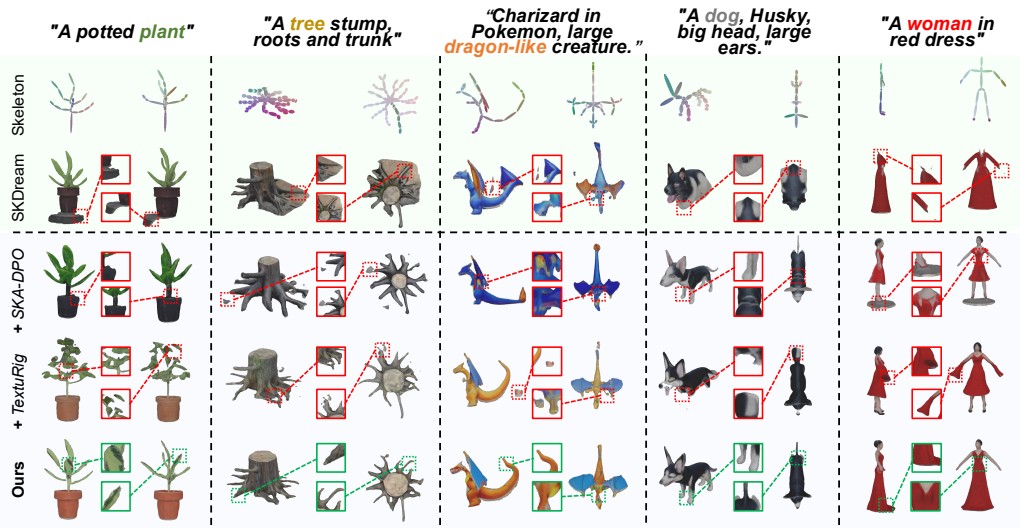

Figure 5: **Qualitative comparison of skeletal-conditioned multi-view generation (§Section 4.3).** Our method produces higher-quality views that more faithfully adhere to the input skeleton. For simplicity, two of the four generated views are shown.

## 4.2 EXPERIMENT SETUP

**Baselines.** We compare the Stroke3D against several methods across two distinct tasks. First, **for the skeleton generation task,** we evaluate our approach against four representative baselines: RigNet (Xu et al., 2020), a classic learning-based method; SKDream (Xu et al., 2025), a non-learning method based on Mean Curvature Flow (Tagliasacchi et al., 2012); MagicArticulate (Song et al., 2025b) and UniRig (Zhang et al., 2025), generating skeleton in autoregressive manner. The second task is **skeleton-to-mesh generation.** Here, we adopt the experimental settings of SKDream to conduct a direct comparison with both SDEdit (Meng et al., 2022) and SKDream itself. It is worth noting that since Stroke3D is the first framework to utilize 2D strokes as input, we deliberately select these state-of-the-art mesh-based rigging methods to ensure a rigorous comparison against the strongest available benchmarks, despite the difference in input modality.

**Evaluation Benchmarks.** We evaluate our generated skeletons and meshes on the MagicArticulate test set and the SKDream evaluation set, respectively. For mesh evaluation, we follow the protocol from SKDream, which uses a test set of 108 samples for comprehensive comparison. Further details on our data processing are provided in the Appendix.

**Evaluation Metrics.** We evaluate Stroke3D's performance using two sets of metrics. To assess skeleton quality, we adopt the Chamfer Distance (CD)-based metrics (Xu et al., 2020), including Joint-to-Joint (CD-J2J), Joint-to-Bone (CD-J2B), and Bone-to-Bone (CD-B2B). These metrics quantify the spatial discrepancy between the predicted and ground-truth skeletons. To evaluate the final generated mesh, we use the SKA score (Xu et al., 2025), which measures the alignment between the input skeleton and the output mesh across multiple views. Following the standard protocol, we perform inference four times for each sample using different random seeds and report the averaged score to account for variance.

Table 3: **Ablation study of preference score margin for SKA-DPO (§Section4.3).** We observe that a margin of 0.1 provides an optimal trade-off across evaluation scores.

| Margin | $\text{Mean}_{Inst.}$ | $\text{Mean}_{Class}$ | Animals | Humans | Plants | Apodes | Bipeds | Wings |
|---|---|---|---|---|---|---|---|---|
| *0.05* | 86.95 | 83.03 | 93.60 | 88.45 | 67.03 | 93.06 | 88.97 | 93.88 |
| *0.15* | **87.84** | 84.21 | **94.08** | 88.66 | 69.89 | 92.73 | **90.42** | 94.25 |
| *0.20* | 86.96 | 83.12 | 93.42 | **88.98** | 66.97 | **94.44** | 87.78 | 93.61 |
| **Ours** (*0.10*) | 87.83 | **84.36** | 93.75 | 88.70 | **70.63** | 93.33 | 89.29 | **94.55** |

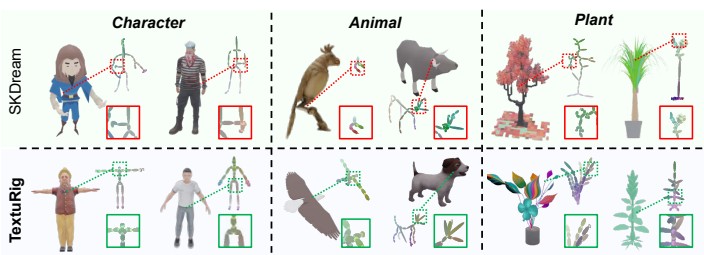
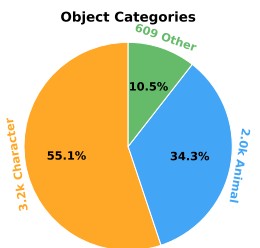

(a) **Skeleton Annotation in mesh generation.** Data in SKdream shows low quality even incomplete (the bird) skeleton with corresponding mesh.

(b) **Breakdown of TextuRig categories.**

Figure 6: **Analysis of the dataset used in mesh generation.**

## 4.3 COMPARISON

**Skeleton Evaluation.** Quantitative results are reported in Tab. 1. Compared with RigNet and SKDream, our method achieves a substantially lower Chamfer Distance across all three metrics, indicating more accurate joint placement and bone connectivity. While MagicArticulate and UniRig shows competitive performance in specific categories such as characters, it performs less consistently on animals and plants. In contrast, Stroke3D achieves balanced improvements across all categories and maintains the lowest overall error. As shown in Fig. 4, existing mesh rigging methods lack explicit structural control, often leading to skeletons with redundant bones in some areas while missing essential ones in others. Stroke3D overcomes this limitation by providing direct structural control, enabling the generation of a skeleton that fully conforms to the user's intent.

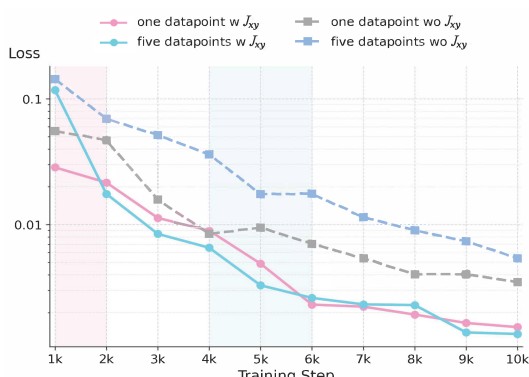

Figure 7: **Ablation study on structural condition (§Section 4.3).** The model converges faster with the structural condition. The pink and blue regions highlight intervals of rapid loss decrease.

**Mesh evaluation.** The results in Tab.2 demonstrate the significant benefit of our proposed TextuRig dataset. By augmenting the training data of the SKDream with TextuRig, we achieve superior skeleton-mesh alignment capabilities compared to the baseline. Notably, our enhanced model surpasses the original SKDream with a 1.9 improvement in $\text{Mean}_{Inst.}$ and a 2.4 increase in $\text{Mean}_{Class}$ score, where these metrics represent the average score per instance and per class, respectively. These results confirm that our TextuRig effectively addresses the data quality limitations of prior work.

As evidenced by the results in Tab. 2, the application of DPO brings a significant boost to our model's performance. We observe a considerable rise in the primary alignment metrics, with the $\text{Mean}_{Inst.}$ and $\text{Mean}_{Class}$ scores reaching **87.84** and **84.21**. Furthermore, our TextuRig and SKA-DPO methods result in more stable mesh generation and exhibit greater fidelity to the conditioning skeleton. As shown in Fig. 5, the vanilla SKDream produces severe geometric artifacts for the *tree stump* and *woman*, whereas our method generates a smooth and stable mesh that accurately conforms to the input pose. As demonstrated in Fig. 8, when subjecting the generated skeleton-mesh pairs to automatic skinning and animation, our meshes maintain robust structural integrity without collapsing, proving their suitability for downstream dynamic applications.

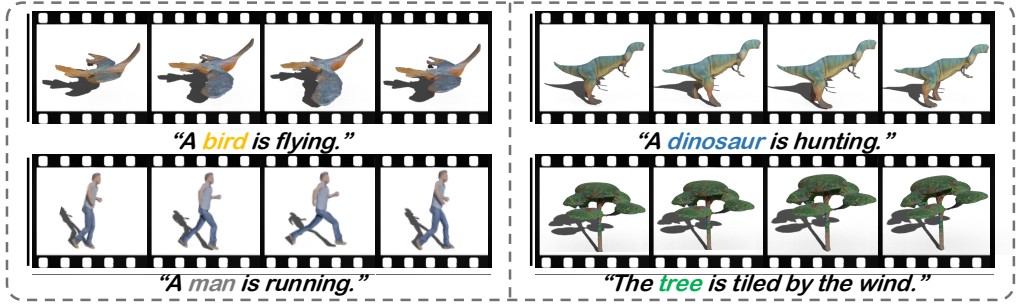

Figure 8: **Qualitative demonstration of animation stability after rigging.** Our method preserves consistent motion dynamics after binding meshes to the skeleton by auto-skinning tools.

## 4.4 ABLATION STUDY

**Ablation study on the impact of the structural condition.** Specifically, we train the model from scratch on a small dataset of either one or five samples, with the goal of forcing overfitting and observing the convergence rate. We compare the performance of models trained with and without this structural condition (denoted as $Jxy$). As shown in the Fig. 7, the results clearly indicate that the absence of the structural condition leads to a markedly slower convergence rate.

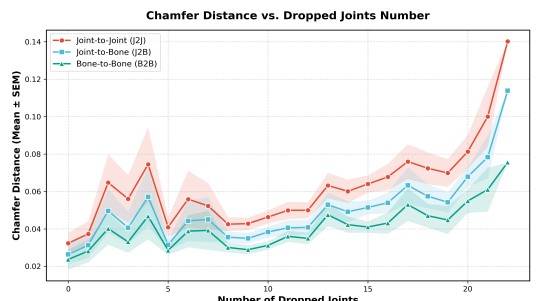

Figure 9: **Sensitivity curve with dropped joints**.

In both the one-datapoint and five-datapoint experiments, the models trained with the $Jxy$ condition (pink and cyan lines) consistently achieve lower loss far more quickly than their counterparts trained without it (grey and blue lines). This finding underscores that the structural prior is crucial for efficient model learning. This observation extends to large-scale training, where we find that the model struggles to converge without this structural condition.

**Ablation study of preference score margin.** We conduct an ablation study to investigate the impact of the preference score margin in SKA-DPO on model performance. This margin is a key hyperparameter that calibrates the desired separation between the scores of chosen and rejected responses. As shown in Tab. 3, a margin of 0.10 achieves an optimal trade-off across the various evaluation scores. Therefore, we adopt a margin of 0.10 as the final setting for our experiments.

**Ablation study on skeleton generation robustness.** We further investigate the sensitivity of our model to input sparsity. As illustrated in Fig. 9, the model maintains high stability and low CD scores when a small number of joints are dropped in skeleton generation. This suggests that our approach is robust to minor occlusions or incomplete user sketches during the inference phase.

## 5 CONCLUSION

In this work, we present Stroke3D, a novel framework for directly generating ready-to-animate 3D assets from intuitive user inputs: 2D strokes and text prompts. Stroke3D introduces a pioneering two-stage pipeline that separates controllable skeleton generation from mesh synthesis, achieving a new level of structural and semantic control. For skeleton generation, our Sk-VAE and Sk-DiT models convert user inputs into high-quality 3D skeletons with precise structural accuracy. Subsequently, our enhanced mesh synthesis, utilizing TextuRig and SKA-DPO, produces detailed, well-aligned geometry and textures. By leveraging this decoupled approach, our method enables the creation of complex rigged assets with high fidelity to the user's vision. Experimental results demonstrate that Stroke3D outperforms existing methods, facilitating a streamlined and intuitive workflow for a wide range of animation applications.

**Acknowledgements.** This work was supported by the "Pioneer" and "Leading Goose" R&D Program of Zhejiang (No. 2024C01161).

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

APPENDICES

## A  DATASET PROCESSING AND ALIGNMENT

This appendix describes the dataset preparation and alignment steps used in our work. Since our framework generates rigged skeletons from 2D strokes and text, it is important to ensure that the dataset we train on is consistent in orientation and structure.

### A.1  DATASET SELECTION

We use **Articulation-XL 2.0** (Song et al., 2025b), which contains about 48K rigged 3D models. To ensure training stability and focus on common subjects, we select the categories *character*, *anthropomorphic*, *animal*, and *plant*, as they typically contain articulated structures suitable for our task. We further filter this subset to include models with a skeleton node count between 0 and 30, which represents the most common range in the dataset. Given that our method performs stroke-to-skeleton generation, which differs from prior mesh-rigging approaches, we apply the same filtering criteria used in our training pipeline to the MagicArticulate test set. This ensures consistency and focuses our evaluation on common object categories.

### A.2  MOTIVATION FOR ALIGNMENT

Although Articulation-XL 2.0 provides a large set of rigged models, their canonical orientations are highly inconsistent. This heterogeneity poses significant challenges for stable training in skeleton generation, as the model may learn with arbitrary orientations instead of learning semantically meaningful skeleton structures. To address this, we enforce a consistent canonical alignment:

- Facing $+Z$ direction (forward axis),
- Head pointing towards $+Y$ direction (up axis),
- Left pointing towards $+X$ direction (right axis),

which implies that the XY-plane corresponds to the **front view**, the XZ-plane to the **top view**, and the YZ-plane to the **side view**.

### A.3  ALIGNMENT STRATEGIES

**Joint-name based method.**

For human-like categories such as *character* and *anthropomorphic*, many skeleton data have joint-level annotations, such as 'head' and 'neck'. When these names contain enough information, we search for keywords such as head, neck, or spine to identify the up-down direction, pairs of joints such as left hand and right hand to identify the left-right direction, and joints such as toe or foot to identify the forward direction. If all three can be confirmed, we directly construct the alignment.

**Structural based method.** If joint names are missing or incomplete, we rely on the geometric structure of the skeleton, especially the symmetry of human legs. We search for pairs of bones that are nearly symmetric with similar lengths and small angular difference. The longest symmetric pair is treated as the legs, and their direction defines the vertical axis. We then check which axis best preserves symmetry when joints are mirrored, and take this as the right axis. The remaining axis is assigned as the forward direction. Finally, if the resulting rotation matrix is not right-handed, we flip the right axis to correct it.

**Principal-direction method.** For categories like *animal* and *plant*, the above two methods often fail because their poses are diverse and less standardized. In this case, we compute main directions directly from the joint positions. We first determine the symmetry axis as before, then estimate the main vertical axis using different priors. For human-like models we apply PCA, assuming the body is tall and narrow. For plants, we assume the dataset root joint is the physical root and use the average bone direction. For animals, we also use the root joint and average bone direction. The remaining axis is set as the forward direction, and we again flip the right axis if necessary.

Table 4: **Ablation study on training data in skeleton generation.** CD scores are calculated over three metrics (CD-J2J, CD-J2B, CD-B2B) across five categories. The lowest and second-lowest scores are shown in bold and underlined, respectively.

| CD Score ↓ | CD-J2J | | | | | CD-J2B | | | | | CD-B2B | | | | |
|---|---|---|---|---|---|---|---|---|---|---|---|---|---|---|---|
| | All | Anthro. | Character | Animal | Plant | All | Anthro. | Character | Animal | Plant | All | Anthro. | Character | Animal | Plant |
| SkDiff-Small | 0.0544 | 0.0556 | 0.0461 | 0.0588 | 0.0543 | 0.0454 | 0.0469 | 0.0378 | 0.0487 | 0.0432 | 0.0400 | 0.0420 | 0.0336 | 0.0412 | 0.0362 |
| SkDiff-Raw | 0.0493 | 0.0498 | 0.0442 | 0.0536 | 0.0407 | 0.0402 | 0.0413 | 0.0367 | **0.0419** | **0.0261** | 0.0346 | 0.0360 | 0.0319 | **0.0348** | 0.0217 |
| SkDiff-NoTag | 0.0524 | 0.0534 | 0.0430 | 0.0587 | 0.0474 | 0.0431 | 0.0442 | 0.0357 | 0.0480 | 0.0316 | 0.0374 | 0.0386 | 0.0324 | 0.0399 | 0.0259 |
| **SkDiff-Full** | **0.0475** | **0.0487** | **0.0390** | 0.0528 | 0.0401 | **0.0389** | **0.0404** | **0.0312** | 0.0432 | 0.0265 | **0.0340** | 0.0358 | **0.0283** | 0.0357 | **0.0214** |

**VLM-assisted method.** For more complex cases, such as animals in unusual poses where heuristics fail, we employ a vision-language model for orientation alignment. We render three orthographic views (XY, YZ, XZ) of the model and then query the VLM (Gemini 2.5 Flash (Team et al., 2023)) to identify which view corresponds to the front, side, and top. Based on the VLM's response, we determine the up, right, and forward axes to construct the rotation matrix. While this method is slower, it offers greater flexibility.

## A.4 Preprocessing for Robustness

To make the above methods more reliable, we apply several preprocessing steps. We first choose the root joint as the one closest on average to all others. Then we split the skeleton into bone segments and assign directions based on distance from the root. Extremely long bones, such as tails, are downweighted so they do not bias the alignment. Finally, we simplify the topology: if many bones share the same starting joint (for example in hands), we average them into one representative segment.

## A.5 Effectiveness of Alignment Methods

The joint-name based method is the most robust, and when sufficient joint names exist it almost guarantees a 100% success rate. The structural based method can correctly recognize most legs, though it may occasionally confuse the front and back, which causes the model to face backward. The principal-direction method works well for most human-like and plant models, but its performance on animals is poor, since different animal categories do not share a common main axis (for example, penguins are upright while crocodiles lie flat). The VLM-assisted method is able to align most animals, but since we render three views without texture, there remain a small portion of animals for which the model cannot judge the views correctly.

In practice, we apply the first three heuristic methods to *character*, *anthropomorphic*, and *plant* models, while we rely on the VLM-assisted method for *animal* models.

## B Skeleton Generation Analysis

In this section, we analyze how training strategies, tagging design, and dataset limitations affect our Skeleton Graph Diffusion (SkDiff) model's performance. We further discuss its generalization ability to unseen concepts and complex sketches, as well as the weaknesses that arise from insufficient data coverage.

### B.1 Training Data and Tagging Strategy

To enhance the semantic understanding of the model, we introduced descriptive tags (e.g., T-pose, symmetry) and viewpoint tags (e.g., front view, side view, top view) during training. Tags were stochastically sampled in the prompt construction process, where one or multiple descriptive tags and one viewpoint tag were optionally appended to the textual input. During validation, all available tags were consistently included. This design encourages the model to learn a more robust association between textual semantics and structural skeleton representation.

As described above, another training strategy we adopt is rotational alignment of the dataset. The motivation is to enhance semantic consistency across training data, thereby reducing unnecessary geometric variance and facilitating model understanding during training

In summary, we conduct a comparison across four training settings: *SkDiff-Small*, trained on a reduced subset of train dataset; *SkDiff-Raw*, trained on the full dataset without rotational alignment;

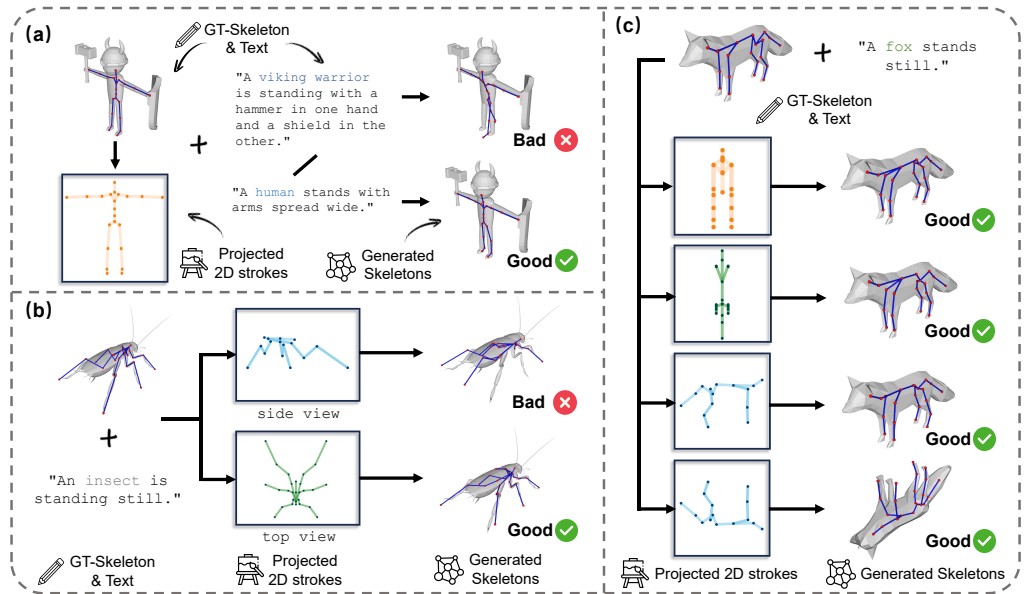

Figure 10: **Qualitative analysis of skeleton generation.** (a) shows the effect of different textual captions on inference results. (b) shows the influence of varying viewpoints. (c) shows the model's ability to generalize across viewpoints and orientations for common categories.

*SkDiff-NoTag*, trained without descriptive or viewpoint tags; and *SkDiff-Full*, trained on the complete dataset with both tagging and alignment. The lowest and second-lowest scores in each column are highlighted in bold and underlined, respectively. The quantitative results are presented in Tab. 4.

We observe that the addition of tags, rotational alignment, and the size of the training data all have a noticeable influence on model performance. Among these factors, the training data size has the most significant impact, with smaller datasets leading to clear degradation. This finding suggests that scaling up the dataset would be the most effective way to further improve skeleton generation quality.

## B.2 GENERALIZATION AND LIMITATIONS

Despite the improvements from tagging and alignment, the model exhibits limitations that are primarily attributable to the scarcity of training data. When confronted with concepts that are rare or absent in the dataset (e.g., samurai or turtle), the model often generates unstable or inconsistent skeletons. The outputs are sensitive to the phrasing of the caption, though carefully designed prompts can mitigate these issues and guide the model towards more reasonable results, as illustrated in case (a) of Fig. 10.

The quality of skeleton generation is also strongly dependent on the information available in the 2D strokes. When strokes provide sufficient structural cues, the model can reconstruct plausible skeletons. Taking Fig. 10 (b) as an example, top-view sketches of insects reveal most joints, enabling reasonable skeleton inference. In contrast, side-view sketches often suffer from overlapping joints, leading to weak constraints and degraded generation. This highlights the challenge of stroke ambiguity in cases where the model lacks strong priors.

For concepts that are well represented in the training data, such as humans and common animals, the model demonstrates good generalization. It is able to generate valid skeletons under novel poses, handle mild noise in sketches (e.g., correcting small rotations), and even interpret inverted inputs by producing logically consistent skeletons, as illustrated in Fig. 10 (c), where the model correctly generates a standing posture for a fox even when the input sketch is vertically flipped. These observations suggest that the model learns transferable priors for frequent categories, though its performance deteriorates quickly for rare or underrepresented concepts.

### B.3 DISCUSSION

Overall, the analysis indicates that the primary bottleneck lies in the dataset. The use of descriptive and viewpoint tags, together with rotational alignment, contributes to better learning of semantic and geometric consistency, but these factors cannot compensate for the limited scale and coverage of the data. Enlarging the dataset and balancing the frequency of rare categories remain the most effective avenues for further improvement. We note the very recent release of Puppeteer Song et al. (2025a), which significantly expands the Articulation-XL dataset utilized in this work. In our future research, we plan to leverage this larger-scale data to mitigate the current data bottleneck. By integrating these expanded resources, we aim to extend Stroke3D into a general-purpose text-to-articulated-asset generation pipeline, capable of handling an even broader spectrum of object categories and complex topological structures beyond the current scope.

## C TRAINING OBJECTIVE OF SK-VAE

We represent a 3D skeleton as an undirected graph $\mathcal{G} = (\mathbf{X}, \mathbf{E})$, where $\mathbf{X} \in \mathbb{R}^{N \times 3}$ represents the 3D coordinates of the $N$ joints, and $\mathbf{E}$ is the set of edges defining the skeleton's topology. The training process is defined by the standard VAE objective. The encoder first maps the input graph to a latent distribution, from which a latent vector $\mathbf{z}$ is sampled. The decoder then reconstructs the coordinates $\mathbf{X}'$ from $\mathbf{z}$, conditioned on the original graph topology $\mathbf{E}$. The model is optimized using the following loss function:

$$\boldsymbol{\mu}, \boldsymbol{\sigma} = \text{Encoder}(\mathbf{X}, \mathbf{E}), \quad \mathbf{z} = \boldsymbol{\mu} + \boldsymbol{\sigma} \odot \boldsymbol{\epsilon}, \text{ where } \boldsymbol{\epsilon} \sim \mathcal{N}(\mathbf{0}, \mathbf{I}), \quad \mathbf{X}' = \text{Decoder}(\mathbf{z}, \mathbf{E})$$
$$\mathcal{L}_{\text{G-VAE}} = \mathcal{L}_{\text{recon}} + \beta \cdot \mathcal{L}_{KL} \tag{3}$$

where $\mathcal{L}_{\text{recon}}$ is the MSE loss between the original coordinates $\mathbf{X}$ and the decoded coordinates $\mathbf{X}'$, and $\mathcal{L}_{KL}$ is the KL divergence between the latent embedding and a standard Gaussian distribution.

The specific formulas for these loss components are: **L2 Loss ($\mathcal{L}_{\text{recon}}$):** This measures the squared Euclidean distance between the original and reconstructed joint coordinates.

$$\mathcal{L}_{\text{recon}} = ||\mathbf{X} - \mathbf{X}'||_2^2$$

**KL Divergence Loss ($\mathcal{L}_{KL}$):** This acts as a regularizer, forcing the distribution of the latent space learned by the encoder to be close to a standard normal distribution. Assuming the latent space has dimension $D$:

$$\mathcal{L}_{KL} = D_{KL}(\mathcal{N}(\boldsymbol{\mu}, \boldsymbol{\sigma}^2 \mathbf{I}) \, || \, \mathcal{N}(\mathbf{0}, \mathbf{I})) = \frac{1}{2} \sum_{i=1}^{D} (\mu_i^2 + \sigma_i^2 - \ln(\sigma_i^2) - 1)$$

## D MORE RESULTS

To further verify the robustness of our method on diverse and complex topologies, we conducted an additional quantitative evaluation on three specific fine-grained categories: Mythical Creatures, Toys, and Weapons. As shown in Table 5, Stroke3D consistently outperforms all baseline methods across all metrics (CD-J2J, CD-J2B, and CD-B2B). Notably, despite Toys and Mythical Creatures being treated as subsets within our broader training categories, our model achieves exceptional precision, reducing the CD-B2B error on Toys to 0.029, significantly surpassing the second-best method, MagicArticulate (0.038). This demonstrates that our topology-driven training strategy effectively generalizes to a wide spectrum of articulated structures, correctly handling the unique skeletal distributions of imaginary creatures and rigid articulated objects alike.

## E ETHICS AND REPRODUCIBILITY STATEMENTS

### E.1 ETHICS STATEMENT

The goal of this research is to democratize the creation of rigged 3D assets, making animation and virtual content creation more accessible to non-professional users.

Table 5: **Quantitative comparison of Chamfer Distance (CD) (§Section4.3).** CD scores are calculated over three metrics (CD-J2J, CD-J2B, CD-B2B) across five categories. The lowest and second-lowest scores are shown in bold and underlined, respectively.

| Method\CD Score ↓ | CD-J2J | | | CD-J2B | | | CD-B2B | | |
|---|---|---|---|---|---|---|---|---|---|
| | Mythi | Toy | Weapon | Mythi | Toy | Weapon | Mythi | Toy | Weapon |
| RigNet [SIGGRAPH 20] | 0.091 | 0.094 | 0.094 | 0.080 | 0.081 | 0.084 | 0.079 | 0.080 | 0.083 |
| SKDream [CVPR25 highlight] | 0.129 | 0.137 | 0.135 | 0.111 | 0.120 | 0.113 | 0.099 | 0.108 | 0.107 |
| MagicArti. [CVPR25] | 0.067 | 0.055 | 0.059 | 0.055 | 0.044 | 0.051 | 0.046 | 0.038 | 0.054 |
| UniRig [SIGGRAPH25] | 0.070 | 0.075 | 0.078 | 0.059 | 0.063 | 0.067 | 0.049 | 0.051 | 0.063 |
| **Ours** | **0.062** | **0.042** | **0.056** | **0.052** | **0.035** | **0.043** | **0.044** | **0.029** | **0.037** |

Table 6: Quantitative comparison of stroke-skeleton alignment (§Section4.3). **We report the 2D Chamfer Distance ($CD_{2D}$) to quantitatively measure the structural alignment between the projected 3D skeleton and the input 2D strokes.** The lowest and second-lowest scores are shown in bold and underlined, respectively.

| Method\2D CD Score ↓ | CD-J2J | | | | CD-J2B | | | | CD-B2B | | | |
|---|---|---|---|---|---|---|---|---|---|---|---|---|
| | All | Character | Animal | Plant | All | Character | Animal | Plant | All | Character | Animal | Plant |
| RigNet [SIGGRAPH 20] | 0.078 | 0.061 | 0.092 | 0.089 | 0.066 | 0.047 | 0.080 | 0.075 | 0.065 | 0.046 | 0.081 | 0.069 |
| SKDream [CVPR25 highlight] | 0.111 | 0.091 | 0.098 | 0.157 | 0.092 | 0.073 | 0.074 | 0.122 | 0.083 | 0.068 | 0.064 | 0.102 |
| MagicArti. [CVPR25] | 0.052 | **0.032** | 0.070 | 0.079 | 0.041 | **0.024** | 0.055 | 0.051 | **0.034** | **0.023** | 0.047 | 0.039 |
| UniRig [SIGGRAPH25] | 0.063 | 0.055 | 0.076 | 0.085 | 0.051 | 0.044 | 0.060 | 0.061 | 0.041 | 0.034 | 0.049 | 0.046 |
| **Ours** | **0.048** | 0.039 | **0.053** | **0.040** | **0.039** | 0.031 | **0.043** | **0.027** | **0.034** | 0.028 | **0.036** | **0.021** |

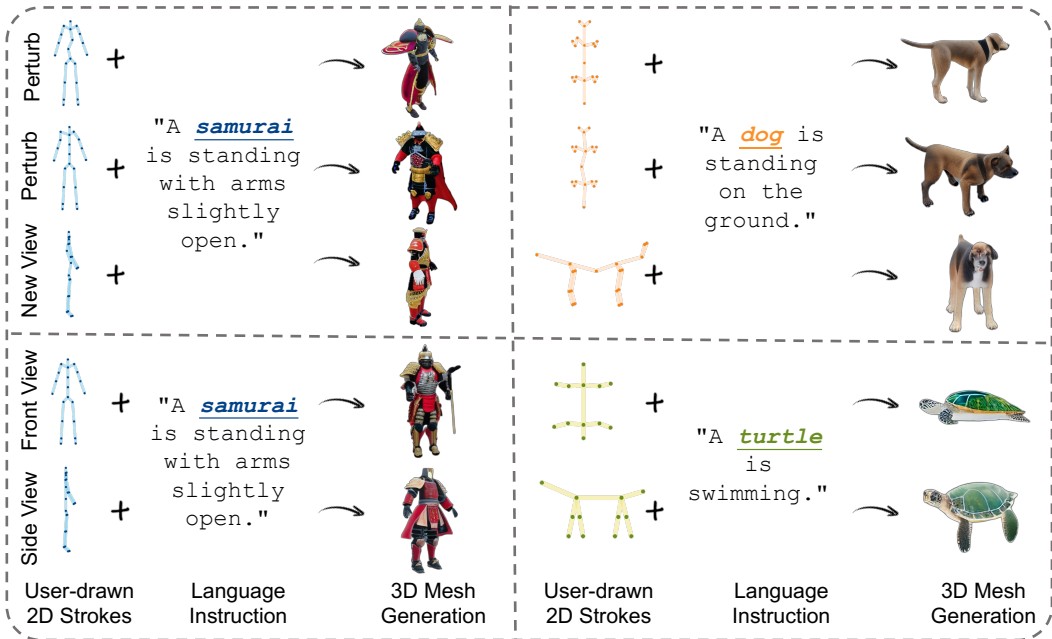

Figure 11: **Qualitative evaluation of generation robustness using our Canvas Tool.** We present results generated from real-world 2D strokes to validate practical usability. The results demonstrate that Stroke3D is (1) robust to input noise, preserving geometric fidelity even with perturbed or jittery strokes; (2) view-invariant, supporting generation from arbitrary camera angles; and (3) generalizable to rare concepts, successfully lifting complex, out-of-distribution subjects (e.g., 'Samurai', 'Turtle') into rigged 3D models.

**Dataset and Licensing:** Our curated dataset, TextuRig, is derived from existing large-scale, publicly available datasets, including Objaverse-XL and the filtered subset from UniRig. We use these assets in a manner consistent with their original licenses and terms of use. The skeleton generation model is trained on the MagicArticulate dataset, which is intended for academic research.

**Potential for Misuse and Bias:** Like other generative models, Stroke3D could potentially be misused to create content that is misleading, biased, or harmful. This is not the intended use of our technology. We also acknowledge that the datasets used for training may contain inherent biases

(e.g., in the distribution of object categories or human representations), which our model may learn and perpetuate. Future work could explore methods to identify and mitigate such biases.

The authors declare no competing interests or conflicts of interest related to this work.

### E.2 REPRODUCIBILITY STATEMENT

We are committed to ensuring the reproducibility of our work. To facilitate this, we provide detailed descriptions of our methodology, data, and experimental setup throughout the paper and its appendices. We plan to release the code for Stroke3D, our pre-trained models (Sk-VAE, Sk-DiT, and the final mesh synthesis model), and the curated TextuRig dataset upon publication of this paper.

## F CONCLUSION

**Limitation and future work.** While our two-stage Stroke3D successfully generates rigged meshes from 2D strokes and text prompts, its performance is constrained by the limited pose variation in the training data. Therefore, our future work will proceed in two key directions. First, we plan to enrich the dataset with more diverse skeletal poses to address the data limitation. Second, we aim to develop an end-to-end network that can produce a rigged mesh directly from a text prompt.

**Usage of LLM.** In this work, we employed a Large Language Model (LLM) to polish the prose and enhance the overall quality of the text.

