# OpenReview forum: "Stroke3D: Lifting 2D strokes into rigged 3D model via latent diffusion models"
_ICLR.cc/2026/Conference — ICLR 2026 Poster_

### Official Review · Reviewer_vd39 · 2025-10-19

**Soundness:** 2
**Presentation:** 2
**Contribution:** 2
**Rating:** 6
**Confidence:** 5

**Summary:**

In this paper, the authors propose Stroke3D, a novel framework that generates mesh-skeleton pairs from 2D strokes and text prompts. The pipeline first uses an Sk-VAE/DiT combination to achieve controllable skeleton generation, conditioned explicitly on the 2D strokes for structure. It then synthesizes the mesh by enhancing a model with the TextuRig dataset and applying the SKA-DPO preference optimization for geometric fidelity.

**Strengths:**

1. The paper tackles an interesting and novel task: generating skeleton-mesh pairs from 2D strokes and text prompts, which has not been explored in prior work.

2. The proposed method demonstrates effective results for skeleton-mesh generation. The two-stage generation pipeline effectively decouples skeleton and mesh generation, allowing each component to be optimized independently while maintaining geometric consistency through the skeleton-conditioned mesh generation stage.

**Weaknesses:**

1. The paper misuses the term "rigged model." In standard 3D graphics terminology, rigging includes both skeleton and skinning weights. Since this work only predicts the skeleton without skinning weights, the authors should use precise terminology such as "skeleton prediction" to avoid misleading readers about the actual technical scope.

2. I do not agree that the dataset is one of the main contributions. (1) It is curated from UniRig rather than being newly collected. (2) Since both SkDream and UniRig source from Objaverse-XL, potential duplicates should be identified and reported. (3) The authors should justify why they did not use MagicArticulate's Articulation-XL, which contains more rigged models. While Articulation-XL lacks textures, these could be obtained from the original Objaverse-XL sources.

3. The method predicts only connected joint positions without their hierarchical tree structure. Prior works (RigNet, UniRig, MagicArticulate) all predict parent-child relationships between joints, which are essential for animation workflows. Without this connectivity information, the output cannot be directly used in standard 3D animation pipelines.

4. The category selection from Articulation-XL (human, animal, plant) lacks justification. Plants are neither a major category in the source dataset nor commonly associated with skeletal rigging needs, unlike toys or articulated objects. The authors should provide selection statistics and explain why underrepresented categories like plants are included while more relevant categories are excluded.

5. The text prompt is only used for skeleton generation, while mesh generation proceeds without textual guidance. This is counterintuitive, as mesh geometry and appearance are more naturally conditioned on text descriptions than skeletal structure.

**Questions:**

1. Missing citation:
Auto-Connect: Connectivity-Preserving RigFormer with Direct Preference Optimization, which also uses DPO for skeleton generation.

2. The current placement of figures, particularly Figure 4 and Figure 5 (Experimental Results), is suboptimal as they appear within the Method section. One suggestion is to rearrange the figures to fit the paper structure.

3. Verify and correct the mark used for the B2B-CD (All) metric in Table 1, as it appears to be incorrect.

4. Include a brief, clear statement in the main paper that you select part of the MagicArticulate Dataset for both training and testing, rather than only mentioning this information in the Appendix.

---

> ### Author Response · Authors · 2025-11-22
> **Answer to weakness1**
>
> We are particularly grateful for your recognition of our core contributions, including **the novelty of our task and the effectiveness of our two-stage pipeline** in decoupling this complex generation problem.
>
> We will address your concerns point by point.
>
> **W1:** Misuse of the concept "rigged model.”
>
> **Answer to W1:**  **We thank the reviewer for this important clarification regarding standard 3D graphics terminology.** We agree that a full "rig" technically includes skinning weights. However, we wish to clarify that our work focuses on the novel and challenging task of generating the **3D skeleton and its corresponding, structurally aligned mesh** directly from user 2D strokes and text.
>
> We made this design choice because the "upstream" generation of a plausible skeleton that matches a mesh is the primary bottleneck in asset creation. Once a high-quality, aligned skeleton and mesh are generated (which our **SKA-DPO** ensures), the subsequent skinning step (assigning weights) is a well-established and mature process.As referenced in our paper, professional tools like Blender provide robust solutions for weight calculation. Therefore, our framework focuses on producing "rig-ready" assets that are immediately compatible with these standard pipelines.
>
> To ensure precise scoping and avoid any potential confusion regarding our technical contributions, we have revised the Introduction **(line 105-107, fig1 caption) (highlighted in blue)** to explicitly state that skinning weights are not a generated output of our model. We clarified that our contribution lies in the controllable generation of the aligned skeleton and mesh, while the computation of skinning weights is delegated to established automated solutions.

---

> > ### Author Response · Authors · 2025-11-22
> > **Answer to weakness3**
> >
> > **W3:** Hierarchical tree structure in skeletons.
> >
> > **Answer to weakness3:** We thank the reviewer for pointing this out. To create the hierarchical tree structure required for animation, we follow the strategy used by SKDream. We compute a **Minimum Spanning Tree (MST)** from the generated skeleton's joints and edges and designate a node (e.g., the one closest to the skeleton's centroid) as the root. This standard process provides the necessary parent-child relationships for animation.
> >
> > As demonstrated in **Figure 1**, this approach is effective and allows our generated assets to be successfully animated (e.g., the running rabbit). And we **add more animation results in Figure 8 in new PDF**, showing that through this strategy that we can lead a robust animation.

---

> > ### Author Response · Authors · 2025-11-22
> > **Answer to weakness5**
> >
> > **W5:** Concern about text condition in mesh generation.
> >
> > **Answer to W5:** We thank the reviewer for this question. In fact, our mesh generation stage, which builds on SKDream, **is conditioned on both skeleton and text**.
> > As detailed in our paper when describing the SKA-DPO stage （line 346-350）, the preference pairs are constructed using both the skeleton $s$ and the **text prompt $c$**. Furthermore, our qualitative results in Figure 5 are all generated with text prompts like "A woman in red dress".
> >
> > However, we realize that this was not made sufficiently clear in the main description of the pipeline. We have revised in **line 289-290** to explicitly state that the text prompt is used as a condition for the mesh synthesis stages.

---

> ### Author Response · Authors · 2025-11-22
> **Answer to weakness2**
>
> **W2:** Concern about dataset contribution and creation.
>
> **Answer to weakness2:** We thank the reviewer for these detailed questions about our dataset. We respectfully disagree that it is not a main contribution, as **its curation was a non-trivial and necessary step to enable our novel task.** We address the reviewer's three points in order:
>
> 1. We respectfully clarify that TextuRig is **a necessary, high-quality resource constructed to resolve a specific deficiency in the field**: the lack of datasets pairing high-quality skeletons with rich textures. This deficiency is a major bottleneck for prior arts. As visually highlighted in Fig. 6 , the training data used by the SKDream exhibits significant quality issues, often containing incomplete skeletons or poor mesh correspondence. Similarly, as noted in the origin paper (line 316-320) datasets like UniRig's, while providing skeletons, are often deficient in corresponding textures. Our process, therefore, used the UniRig-filtered subset only as a starting point to identify a list of relevant models. **We did not use their processed results; instead, we re-downloaded the original assets.** We then re-processed these assets, introducing our own rigorous filtering step to isolate models that possess high-quality texture information (e.g., "vertex colors or a texture map") —a key feature many of the original assets lacked. Furthermore, we generated new, rich captions for these models.
>
> 2. Although both SKDream and UniRig ultimately trace back to Objaverse-XL, the specific meshes used **are distinct**. And we can confirm there is **no duplication of the (skeleton, mesh) training pairs** between our UniRig and SKDream's dataset. The reason is that both the skeletons and the mesh subsets are fundamentally different: SKDream generates its own skeletons from meshes using Mean Curvature Flow (MCF).Unirig uses the original, human-annotated skeletons that were provided with the models in Objaverse-XL.
>
> 3. 3. We chose not to use Articulation-XL due to its data source heterogeneity and the unreliable availability of textures. As stated in its public repository, the Articulation-XL dataset aggregates models from diverse sources, including both Objaverse-XL and various GitHub repositories.  The processed files released by Articulation-XL lack texture information. **More importantly, due to the varied sources (especially from GitHub), there was no guarantee that the original assets would even contain textures if we attempted to trace them back.** Therefore, we chose the UniRig-filtered subset as our starting point.
>
> However, we appreciate the reviewer's constructive feedback, which has helped us improve the clarity of our contributions. **In Lines 96-98, 223-225, and 288-290 (highlight in blue),** we have expanded the description of our data processing pipeline to clarify the rigorous filtering and re-captioning steps, emphasizing the critical necessity of TextuRig beyond simple dataset extension. We believe these clarifications will help future readers better appreciate the foundational advancements presented in this work.

---

> ### Author Response · Authors · 2025-11-22
> **Answer to weakness4 and question4**
>
> **W4: Justification for category selection. & Q4 Brief statement for dataset selection.**
>
> **Answer to W4 & Q4**: We thank the reviewer for this detailed question regarding our category selection. We chose these categories based on three primary factors: **fair benchmarking alignment and broad label coverage, the necessity of explicit structural control for generation.** And we will introduce a brief dataset selection.  We address these points below:
>
> 1. **Alignment with Established Methods:** Our selection of 'Human', 'Animal', and 'Plant' was primarily driven by the need to ensure a strict, fair comparison with one of our primary baseline, SKDream. The established SKDream benchmark explicitly evaluates performance across these exact three categories ('Character', 'Animal', 'Plant'). While the Articulate-XL dataset contains various fine-grained tags, objects often carry multiple labels. To ensure a unified evaluation protocol, we adopted the high-level taxonomy used. Excluding plants would have prevented a valid head-to-head comparison with the baseline. Regarding the concern about excluding 'toys' or 'articulated objects', we clarify that these are not excluded but subsumed under broader semantic labels. In the MagicArticulate dataset, a "toy" is frequently cross-listed as a "Character" or "Animal" depending on its morphology.
>
> 2. **Skeletal Rigging as an Intuitive Proxy for Explicit Shape Control.** We respectfully offer a different perspective on the reviewer's comment that plants lack rigging needs. Users often have specific intent for the shape of an object (e.g., a tree bending in a specific direction or a flower with a specific bloom structure). Generating such objects without explicit guidance is difficult. Our framework **uses the skeleton as a low-dimensional proxy for this geometry, allowing users to intuitively define the object's structure via strokes.**  Furthermore, in modern production, rigged vegetation is essential for dynamic environmental effects, such as simulating realistic wind sway or interaction with characters.
>
> 3. **Dataset Selection.** We explicitly state in **line 318-323 (highlight in blue) of revised PDF** that we filter the MagicArticulate (Articulation-XL) dataset to retain skeletons with 0-30 nodes. We selected this range because it is **highly representative, covering the vast majority of common articulated objects found in the real world and standard graphics applications.** We found that dataset distribution does not scale uniformly with node count. Specifically, models with >30 nodes are disproportionately dominated by "Characters" featuring dense, specialized rigs for non-structural elements (e.g., hair dynamics, clothing simulation). In contrast, the 0-30 range is structurally representative, capturing the vast majority of "Animal" species and standard "Human" skeletal topologies. This aligns with established pose estimation standards (e.g., COCO Keypoints, which utilize 17 points for humans), ensuring our model focuses on learning fundamental body structures rather than simulation artifacts.
>
> 4. **Data label process.** We first employed a Vision-Language Model (VLM) to generate descriptive captions for each mesh, ensuring semantic accuracy beyond the raw dataset tags. By combining these VLM captions with the original Articulation-XL metadata, we analyzed the distribution of the filtered subset. Based on this analysis, we consolidated the data into three structural categories: Character (44%), Animal (50%), and Plant (6%). Crucially, regarding the reviewer's concern about "Toys," our analysis of raw labels shows that items explicitly tagged as "Toy" account for approximately 4.6%. **However, these assets are not excluded; instead, they are fully integrated into the 'Character' and 'Animal' categories based on their skeletal topology (e.g., a toy robot is structurally a character).**
>
> **Future Work with Puppeteer:** We acknowledge that data scale remains a limiting factor for broader categories. We have updated our discussion to note that the very recent release of Puppeteer has significantly expanded the Articulation-XL dataset. We plan to leverage this larger-scale data to extend our framework into a general-purpose text-to-articulated-asset generation pipeline in future work.
>
> We sincerely thank the reviewer for constructive suggestions, which have prompted us to strengthen our evaluation and clarity. To demonstrate the versatility of our method beyond the primary categories, we have added new quantitative evaluations on 'Mythological Creatures', 'Toys', and general 'Creatures' in our test set, detailed in **Appendix D (Table 5).** Furthermore, we have explicitly incorporated the skeleton dataset filtering criteria into the main text in **Lines 318-323)** and expanded the discussion on future directions in the Conclusion **(Lines 920-928)**. All significant revisions in the updated manuscript are highlighted in **blue** for convenience.

---

> ### Author Response · Authors · 2025-11-22
> **Answer to question1**
>
> **Q1:** Missing citation.
>
> **Answer to Q1:** We have made our best efforts to ensure the completeness and comprehensiveness of the references in our paper. However, due to the rapid development of this field, it is challenging to cover all emerging works. **We sincerely thank the reviewer for bringing this relevant work to our attention. We have updated the manuscript to include the citation for Auto-Connect in Lines 185-186 (highlighted in blue) to further contextualize our DPO strategy within the latest literature.**

---

> ### Author Response · Authors · 2025-11-23
> **Answer to question2**
>
> **Q2:** The current placement of figures, particularly Figure 4 and Figure 5 (Experimental Results), is suboptimal as they appear within the Method section. One suggestion is to rearrange the figures to fit the paper structure.
>
> **Answer to Q2:** We thank the reviewer for this helpful suggestion regarding the paper's layout. We agree that placing experimental figures closer to the relevant quantitative analysis improves the flow and readability of the paper. And we have changed our layout as you suggested.

---

> ### Author Response · Authors · 2025-11-23
> **Answer for question3**
>
> **Q3:** Verify and correct the mark used for the B2B-CD (All) metric in Table 1, as it appears to be incorrect.
>
> **Answer to Q3:** We thank the reviewer for their careful feedback. This was indeed a clerical error on our part, and we have corrected it in the revised manuscript.

---

### Official Review · Reviewer_6nvo · 2025-10-29

**Soundness:** 3
**Presentation:** 3
**Contribution:** 2
**Rating:** 2
**Confidence:** 4

**Summary:**

The paper introduces Stroke3D, a novel framework that generates rigged 3D meshes from user-drawn 2D strokes and text prompts. Unlike prior methods that generate a 3D mesh first and then rig it, Stroke3D adopts a skeleton-first generation pipeline. Experiments on MagicArticulate and SKDream benchmarks show that Stroke3D outperforms baselines such as RigNet, UniRig, and SKDream, achieving better Chamfer Distance and SKA scores. The system demonstrates both structural control and high visual quality, enabling intuitive rigged 3D asset generation.

**Strengths:**

1. First framework enabling rigged 3D generation directly from 2D strokes and text.
2. Outperforms baselines such as RigNet, UniRig, and SKDream, achieving better Chamfer Distance and SKA scores.
3. Introduce TextuRig, a dataset of textured and rigged 3D meshes with captions

**Weaknesses:**

1. Other pipelines generate skeletons from 3D meshes, while this paper generates skeletons directly from 2D strokes. Therefore, comparing skeleton quality metrics between these methods is not entirely fair, since the input modalities differ significantly in both information richness and structural constraints.

2. The contributions of this paper are relatively limited. It mainly proposes a model that generates 3D skeletons from 2D strokes and introduces a modest extension of the SKDream dataset, rather than delivering fundamentally new insights or large-scale advancements to the field.

**Questions:**

1.How is the camera viewpoint determined when projecting the 3D skeleton into 2D strokes?

2.What are the specific design details of the Skeletal Graph Variational Autoencoder (Sk-VAE)? According to the paper, the VAE decoder requires the set of edges to reconstruct the skeleton. So how are these edges obtained during the generation process?

---

> ### Author Response · Authors · 2025-11-22
> **Answer to weakness1**
>
> We are particularly grateful for your recognition of **our novelty as the first stroke-and-text guided framework, our strong quantitative performance, and the technical contribution** of our TextuRig dataset.
>
> We will address your concerns point by point.
>
> **W1:** Unfair comparisons with other skeleton generators.
>
> **Answer to W1:** We thank the reviewer for this insightful comment regarding the comparison fairness. We respectfully argue that **we have strived to ensure this comparison is as fair as possible, and it is essential to demonstrate the core contribution of Stroke3D.** We address this issue from three perspectives: the pioneering nature of our 2D stroke-based paradigm, the rationale for selecting state-of-the-art baselines , and the validity of adopting standard community metrics.
> 1. **Pioneering 2D Structural Control to Resolve Skeleton Generation Ambiguity.** Stroke3D is **the first framework to introduce explicit 2D structural control into 3D skeleton generation** (via user-drawn strokes), to solve the "unpredictable results" problem inherent in traditional mesh-to-skeleton approaches. As noted in our introduction (Lines 47-51 in origin version PDF), existing methods (e.g., MagicArticulate, UniRig) infer skeletons from finished meshes. This often leads to logical errors(in Fig. 4)—generating bones where none are needed or missing critical joints. **Crucially, due to the pioneering nature of our work, there are simply no existing methods that share the same input modality (2D strokes to 3D skeleton). Consequently, we were compelled to compare against mesh-based methods as the only viable alternative to establish a quality benchmark.** This comparison was not a choice of convenience, but a necessity to validate our method's utility in the absence of direct counterparts.
>
> 2. **Careful Selection of Strong Baselines for Rigorous Benchmarking.** Given the necessity to benchmark against mesh-based methods, we carefully selected MagicArticulate and UniRig as baselines **because they represent the current state-of-the-art (SOTA), utilizing large-scale datasets for high generalization**. The reviewer correctly notes that the baselines **utilize 3D meshes as input, which are inherently more informative and contain complete 3D geometric cues compared to our sparse 2D strokes.** Despite this disadvantage in input information richness, Stroke3D achieves comparable or superior performance. As shown in Table 1, our method achieves the lowest Chamfer Distance.
>
> 3. **Adoption of Standard Community Metrics Ensures Objective Output Assessment.** To ensure the validity of our experiments, we strictly adhered to the **standard evaluation protocols in this domain.** We utilized **Chamfer Distance (CD)** metrics, which are the **universally accepted standards for quantifying the spatial discrepancy between predicted and ground-truth skeletons.** Regardless of the input modality, the ultimate goal for a user is to obtain a high-quality, anatomically plausible skeleton. By benchmarking on these standard metrics, we provide a direct measure of the *usability* of our generated assets compared to existing tools.
>
> However, we agree with the reviewer that the distinction in input modalities and the rationale for our experimental design should be explicitly clarified to avoid confusion. We have revised the manuscript **(highlighted in blue)** to address this. In Lines **74-75**, we explicitly highlight the pioneering nature of our "Skeleton-First" paradigm, emphasizing how it empowers user intent where traditional methods fail. In **Lines 380-382**, we have added a detailed justification for selecting these baselines, clarifying that we deliberately benchmark against the strongest available mesh-based methods to ensure a rigorous evaluation despite the difference in input information.

---

> ### Author Response · Authors · 2025-11-22
> **Answer to weakness2**
>
> **W2:** Limited contribution.
>
> **Answer to W2**: We thank the reviewer for the feedback. We respectfully wish to clarify that our contributions are not limited; rather, we represent **fundamental contributions in algorithmic design, critical data curation, and optimization strategy that unlock a new task previously unattainable.** We address these three core contributions below:
>
> 1. **Fundamental Contribution in algorithmic design:** our work establishes a **fundamentally new paradigm** for rigged asset generation. We are the first to design and apply a latent graph diffusion model (Sk-DiT) to synthesize complete 3D skeletons conditioned on 2D strokes and text. This is not merely a new model architecture, but a fundamental shift designed to resolve the "unpredictable results" caused by the lack of explicit structural control in prior methods. This innovation empowers non-professional users to effortlessly generate ready-to-animate 3D assets with their intent.
>
> 2. **Contribution in Data Curation (Beyond "Extension"):**  We respectfully clarify that **TextuRig is a necessary, high-quality resource constructed to resolve a specific deficiency in the field:** the lack of datasets pairing high-quality skeletons with rich textures. This deficiency is a major bottleneck for prior arts. **As visually highlighted in Fig. 6,** the training data used by the SKDream exhibits significant quality issues, often containing incomplete skeletons or poor mesh correspondence. Similarly, as noted in the **origin paper (line 316-320)** datasets like UniRig's, while providing skeletons, are often deficient in corresponding textures. Our process, therefore, used the UniRig-filtered subset only as a starting point to identify a list of relevant models. **We did not use their processed results; instead, we re-downloaded the original assets.** We then re-processed these assets, introducing our own rigorous filtering step to isolate models that possess high-quality texture information (e.g., "vertex colors or a texture map") —a key feature many of the original assets lacked. Furthermore, we generated new, rich captions for these models.
>
> 3. **Contribution in Optimization:** Our contribution lies in its specific design for the task of skeleton-conditioned mesh generation. To our knowledge, **no prior work** has applied preference optimization to this specific problem. Our key Optimization contribution is that we term SKA-DPO: we propose the innovative use of the SKA Score as the preference signal. This specific choice directly optimizes the model for the crucial goal of structural coherence. As quantitatively demonstrated by our ablation studies (Table 2), this strategy is highly effective, "significantly enhances the final mesh's geometric quality" , and rectifies key weaknesses in the baseline model.
>
> However, we realize that the distinct value and non-incremental nature of these contributions could be articulated more clearly for future readers. We have revised the manuscript **(highlighted in blue)** to explicitly emphasize these points. Lines **74-75**, we explicitly highlight the pioneering nature of our "Skeleton-First" paradigm that empowers user intent. In **Lines 96-99, 223-225, and 288-290**, we have expanded the description of our data processing pipeline to clarify the rigorous filtering and re-captioning steps, emphasizing the critical necessity of TextuRig beyond simple dataset extension. We believe these clarifications will help future readers better appreciate the foundational advancements presented in this work.

---

> ### Author Response · Authors · 2025-11-22
> **Answer to question1**
>
> **Q1:** How is the camera viewpoint determined when projecting the 3D skeleton into 2D strokes?
>
> **Answer to Q1:** We thank the reviewer for this clarifying question, which allows us to be more precise about our data generation process. However, our method does not utilize a perspective camera model (which would involve a defined focal length and camera position). Instead, we employ standard **orthographic projection** to simulate the user's drawing canvas. To simulate different "viewpoints" during training, we do not move a virtual camera; rather, we transform the object itself. Crucially, since **we employ random 3D rotations and joint perturbation for data augmentation during this process, our model demonstrates strong robustness.** The process is as follows:
>
> 1. We take the 3D ground-truth skeleton.
> 2. We apply a **3D rotation and joint perturbation** to the entire skeleton object for data augmentation.
> 3. We then project its 3D joint coordinates orthographically (i.e., we simply take the X and Y coordinates) onto the 2D plane.
>
> We appreciate the reviewer's suggestion to clarify our data generation process. We have revised the "Data Processing" paragraph in Section 4.2 **(Lines 455-456, highlighted in blue)** to explicitly describe this "Rotation-then-Projection" pipeline and further highlight the model's robustness to varying user viewpoints. Furthermore, we have expanded experiment in **Appendix Fig11** to further highlight the model's robustness to varying user viewpoints and joint pertubations.

---

> ### Author Response · Authors · 2025-11-22
> **Answer to question2**
>
> **Q2:** What are the specific design details of the Skeletal Graph Variational Autoencoder (Sk-VAE)? According to the paper, the VAE decoder requires the set of edges to reconstruct the skeleton. So how are these edges obtained during the generation process?
>
> **Answer to Q2:** We thank the reviewer for this question. We will answer it by addressing two key aspects: (1) the pioneering task-oriented design of our Sk-VAE, and (2) how our canvas tool is explicitly designed to capture the edge topology ($E$) for vae directly from the user's input.
>
> 1. **Pioneering Design of the First Skeletal Graph VAE:** Our Sk-VAE is not a standard graph autoencoder; it is a **pioneering design specifically engineered to enable the first latent diffusion framework for 3D skeletons.** Inspired by the success of Latent Diffusion Models (LDMs) in image generation, we recognized the need for a robust, continuous latent space to handle 3D skeletal data. However, unlike grid-based images, skeletons are graph-structured. To solve this unique challenge, we designed the Skeletal Graph VAE (Sk-VAE) to explicitly encode this graph structure. As detailed in Section 3.2, we employ GCN and TransformerConv layers to effectively aggregate features and update representations between connected nodes.
>
> 2. **On How the Decoder Obtains the Edges (E):** The edges (topology $E$) are obtained directly from the user's 2D stroke input. **Our canvas tool is motivated by professional 3D software (like Blender),** where skeletons are defined by joints and connected line segments ("bones"). We replicated this structure: our tool guides users to **draw with 2D connected line strokes rather than free-form sketches.** This design choice inherently aligns the structure of "real" user inputs with our graph-based training data, significantly narrowing the perceived domain gap from the outset. Through this tool design, our training process was designed to anticipate real-world imprecision. This user-drawn stroke(2D graph) defines the edge topology $E$ from the very beginning. Therefore, the decoder knows the edge set because the user provided it as part of the input.
>
> We thank the reviewer for their valuable feedback, which helped us identify that the role of the user-defined topology needed to be stated more explicitly. We have revised canvas design in line **362-370 (highlight in blue).**

---

### Official Review · Reviewer_X4bw · 2025-10-31

**Soundness:** 2
**Presentation:** 2
**Contribution:** 2
**Rating:** 6
**Confidence:** 4

**Summary:**

This paper proposes a novel framework for synthesizing rigged 3D skeletons from 2D drawn strokes and text prompts. The authors first curate a dataset from existing open-source 3D rigging collections, then train skeleton VAE and diffusion models to enable generation conditioned on 2D strokes. To further enhance performance, the paper introduces a skeleton-mesh alignment process guided by an optimization step, improving the model's fidelity. Experiments demonstrate that the proposed method outperforms existing approaches.

**Strengths:**

1. The paper addresses an interesting and novel task: generating 3D articulated objects from a combination of 2D strokes and text descriptions. The problem itself is well-motivated and interesting.
2. The work improves upon existing baselines in several areas, including dataset curation, VAE design, and a post-hoc optimization stage. This results in a stronger baseline for future research in rigged 3D generation.
3. The paper is well-written, clearly structured, and easy to follow.

**Weaknesses:**

1. My main concern revolves around the paper's technical novelty. The proposed method appears to be a successful combination of existing technologies:
* A data curation step that refines datasets from prior work (Unrig, MagicArticulate).
* Slight modifications to the existing Sk-dream model architecture.
*  The application of a post-training optimization phase, which is a common technique for refinement.
Consequently, the contribution seems to lie more in effective engineering and integration than in providing new fundamental insights.

2. The experimental evaluation is not sufficient. A key claim is the ability to generate rigged objects from 2D sketches, yet there is no quantitative metric to evaluate the consistency between the input 2D sketch and the generated 3D object. I suggest that the authors include such a metric (e.g., projection-based distance) to substantiate their claims.

3. The process for creating the training data is not sufficiently detailed. Specifically, the paper should clarify how the paired 2D drafts and their corresponding 3D ground-truth (GT) models were generated. This information is crucial for reproducibility.

**Questions:**

A potential limitation of the method appears to be its robustness to imperfect user inputs. The paper primarily showcases results with clean, well-defined 2D drafts. However, in real-world scenarios, user-drawn strokes are often noisy, shaky, or not perfectly straight. How would the model perform under such conditions? I would encourage the authors to provide an analysis or at least a qualitative study on the model's sensitivity to variations in input stroke quality.

---

> ### Author Response · Authors · 2025-11-22
> **Answer to weakness1**
>
> We sincerely appreciate your recognition of **the motivation of our work, strength of our technical contribution and our clear presentation and writing**.  Thank you very much for your valuable comments!
>
> We will address all the concerns point by point.
>
> **W1:** Concern of technical novelty
>
> **Answer for weakness1:** We respectfully wish to clarify that our technical novelty is not limited; rather, we represent **fundamental novelty in algorithmic design, critical data curation, and optimization strategy that unlock a new task** previously unattainable. We address these three core novelty below:
>
> 1. **Fundamental novelty in algorithmic design:** our work establishes a **fundamentally new paradigm for rigged asset generation**. As we introduce in **Line 106-107 of the origin version PDF**, we are the first to design and apply a latent graph diffusion model (Sk-DiT) to synthesize complete 3D skeletons conditioned on 2D strokes and text. This is not merely a new model architecture, but a **fundamental shift** designed to resolve the "unpredictable results" caused by the lack of explicit structural control in prior methods. This innovation empowers non-professional users to effortlessly generate ready-to-animate 3D assets with their intent.
>
>     **For Mesh Synthesis**, our contribution here was not to reinvent the architecture, **but rather to solve baseline’s known weaknesses** through two other key contributions: our novel TextuRig dataset, which provides essential high-quality data , and our SKA-DPO optimization strategy, which effectively solves its structural alignment weaknesses. As our ablation study in Table 2 clearly demonstrates, these contributions together dramatically improve the baseline's performance and alignment.
>
> 2. **Novelty in Data Curation (Beyond "Refinement")**:  We respectfully clarify that TextuRig is a necessary, high-quality resource constructed to resolve a **specific deficiency in the field:** the lack of datasets pairing high-quality skeletons with rich textures. This deficiency is a major bottleneck for prior arts. As visually **highlighted in Fig. 6 in origin PDF**, the training data used by the SKDream exhibits significant quality issues, often containing incomplete skeletons or poor mesh correspondence. Similarly, as noted in the **origin paper (line 316-320)** datasets like UniRig's, while providing skeletons, are often deficient in corresponding textures. Our process, therefore, used the UniRig-filtered subset only as a starting point to identify a list of relevant models. **We did not use their processed results; instead, we re-downloaded the original assets.** We then re-processed these assets, introducing our own rigorous filtering step to isolate models that possess high-quality texture information (e.g., "vertex colors or a texture map") —a key feature many of the original assets lacked. Furthermore, we generated new, rich captions for these models.
> 3. **Novelty in Optimization**: Our contribution lies in its **specific design for the task** of skeleton-conditioned mesh generation. To our knowledge, **no prior work has applied preference optimization to this specific problem**. Our key Optimization contribution is that we term SKA-DPO: we propose the innovative use of the SKA Score as the preference signal. This specific choice directly optimizes the model for the crucial goal of structural coherence. As quantitatively demonstrated by **our ablation studies (Table 2)**, this strategy is highly effective, "significantly enhances the final mesh's geometric quality" , and rectifies key weaknesses in the baseline model.
>
> However, we realize that the distinct value and non-incremental nature of these contributions could be articulated more clearly for future readers. We have revised the manuscript **(highlighted in blue)** to explicitly emphasize these points. **In lines 74-75**, we explicitly highlight the pioneering nature of our "Skeleton-First" paradigm that empowers user intent. **In Lines 95-99, 223-225, and 288-290**, we have expanded the description of our data processing pipeline to clarify the rigorous filtering and re-captioning steps, emphasizing the critical necessity of TextuRig beyond simple dataset extension. We believe these clarifications will help future readers better appreciate the foundational advancements presented in this work.

---

> ### Author Response · Authors · 2025-11-22
> **Answer to weakness2**
>
> **W2:** Concern about alignment between 2D stroke and 3D mesh.
>
> **Answer to W2:** We thank the reviewer for this highly constructive suggestion. We agree that quantifying **the consistency between the input 2D stroke and the final 3D mesh is crucial.** To rigorously demonstrate this, we validate the "Chain of Consistency" through two logical steps: (1) alignment between Stroke and Skeleton, and (2) alignment between Skeleton and Mesh.
>
> 1. Quantitative Alignment between 2D Stroke and 3D Skeleton. To directly address the reviewer's request, we have implemented a new "2D Chamfer Distance" evaluation **in Table 6 (Appendix of new PDF)**. We project the generated 3D skeleton joints back onto the 2D canvas plane using the same orthographic projection parameters used during inference. We then calculate the Chamfer Distance (CD) between the set of points in the projected 3D skeleton and the set of points in the user's original 2D input strokes.
>
> 2. **Quantitative Alignment between Skeleton and Mesh via SKA Score.**
> Once the skeleton is aligned with the stroke, the remaining task is to ensure the mesh aligns with the skeleton. We respectfully remind the reviewer that the SKA Score we utilized is inherently a "multi-view projection-based metric." As detailed in our paper, SKA measures the alignment between the projected skeleton and the rendered mesh across multiple views. As shown in Table 2, our method achieves a high SKA score (87.83 Mean Inst.), significantly outperforming the baseline. This confirms that the final mesh faithfully wraps around the 3D skeleton.
>
> Since Step 1 proves the 3D skeleton aligns with the 2D stroke, and Step 2 proves the 3D mesh aligns with the 3D skeleton, we quantitatively demonstrate that the **final 3D mesh is structurally consistent with the initial 2D stroke input.**
>
> However, to further enhance the readability and clarity of our manuscript, we have added this new "2D Chamfer Distance" experiment and analysis to **Appendix  (Table 6)** to further substantiate our claim of structural controllability. And we have added the mesh evaluation process more correctly in **Line 431-433 (highlight in blue).**

---

> ### Author Response · Authors · 2025-11-22
> **Answer to weakness3**
>
> **W3:** Concern about 2d stroke-3d mesh dataset preparation.
>
> **Answer to Weakness3:** We thank the reviewer for this important question. Our method **does not use a single dataset** of (2d stroke, 3d GT mesh) pairs. Instead, we have **two separate data pipelines** for our two-stage model: And we will answer this question through our two-stage data pipeline.
>
> 1. **The skeleton generation stage (2D Stroke to 3D Skeleton)** is trained on (Simulated 2D Stroke, 3D GT Skeleton, Text) triplets. We create this data, as detailed in Section 3.3 and Section 4.2 in origin PDF, by taking 3D ground-truth skeletons (from MagicArticulate), creating 2D projections, and applying perturbations to "mimic the imprecision inherent in hand-drawn strokes".
>
> 2. **The mesh generation stage (3D Skeleton to 3D Mesh)** is subsequently trained on a separate dataset of (3D GT Skeleton, 3D GT Mesh, Text) triplets. This dataset combines the SKDream dataset with our curated TextuRig dataset, which provides the necessary (skeleton, textured-mesh) pairs. We hope this direct explanation of our two distinct data pipelines fully resolves the concern.
>
> To further enhance the readability and clearly distinguish the data sources for each stage, we have revised the Experiment section **(highlighted in blue)**. In **Lines 318-323**, we explicitly state that the skeleton generation stage is trained on MagicArticulate, emphasizing its status as the largest recent rigging dataset to ensure robust structural priors. Furthermore, in **Lines 349-351**, we clarify that the mesh synthesis stage utilizes the SKDream dataset augmented with our curated TextuRig to ensure high-quality texture alignment.

---

> ### Author Response · Authors · 2025-11-22
> **Answer to question1**
>
> **Q1:** Robustness on user-drawn strokes.
>
> **Answer to Q1:** We thank the reviewer for highlighting these important challenges. We would like to clarify that our framework possesses **robustness to real-world drawings.**  We will address these points in two primary ways: first, through the design of our canvas tool which inherently bridges the domain gap, and second, through qualitative examples and addition experiments in the paper that show our model handling varied and imperfect user inputs.
>
> 1. **Bridging the Domain Gap via Constrained Tool Design.** As seen in Figure 1 of our original paper, all generation examples (including the human, dinosaur, bird, and tree) were **created by humans using our provided canvas tool, not by simulated data.** Our canvas tool is **motivated by professional 3D software (like Blender),** where skeletons are defined by joints and connected line segments ("bones"). We replicated this structure: our tool guides users to **draw with 2D connected line strokes rather than free-form sketches.** This design choice inherently aligns the structure of "real" user inputs with our graph-based training data, significantly narrowing the perceived domain gap from the outset. Through this tool design, our training process was designed to anticipate real-world imprecision. As described in Section 3.3, we simulate stroke data by "apply perturbations to 2D projections, which **mimics the imprecision inherent in hand-drawn strokes"**. This step explicitly trains the model to handle the uncertainty of human drawing.
>
> 2. **Empirical Evidence:** We would like to highlight that our original analysis provided evidence for the model's ability to handle stroke ambiguity and varied viewpoints. As discussed in Appendix B.2 and shown in **Figure 8(c), the 'fox'** example demonstrates strong generalization. The model successfully generates a logically consistent skeleton even from a vertically flipped stroke, proving it learns a robust prior that generalizes beyond simple projections and can handle significant structural ambiguity. Furthermore, Figure 1 demonstrates practical application on different views, such as a human from a side-view ('A man is running...') and a man from a frontal-view ('A man is standing.').
>
> However, we agree that explicit clarification and further empirical validation would strengthen the paper. We have added a detailed description of our canvas tool's design in **Lines 362-370 (highlighted in blue)** to clarify how it inherently aligns user inputs with skeletal graph topology . Additionally, we have revised the caption of **Figure 1 (highlighted in blue)** to explicitly state that these results are practical applications generated from real human-drawn strokes. Furthermore, to rigorously demonstrate the model's robustness against input perturbations and varying viewpoints (rotations), we are conducting additional experiments. We present these new qualitative visualizations in **Appendix Fig 11** of the revised manuscript.

---

### Official Review · Reviewer_zQuz · 2025-11-05

**Soundness:** 3
**Presentation:** 3
**Contribution:** 3
**Rating:** 4
**Confidence:** 3

**Summary:**

The paper presents Stroke3D, a skeleton-first pipeline that produces rigged 3D meshes from user-drawn 2D strokes plus a text prompt. The method learns a latent space for skeletal graphs with a Skeletal Graph VAE (Sk-VAE) and performs denoising in that space using a Skeletal Graph Diffusion Transformer (Sk-DiT) conditioned on text and stroke features; the decoded skeleton then conditions a mesh generator. A second stage improves skeleton-to-mesh synthesis by curating TextuRig (a textured, rigged subset with captions) and applying skeleton–mesh-alignment-guided Direct Preference Optimization (SKA-DPO). On MagicArticulate (skeletons) and SKDream (meshes), the approach lowers Chamfer distance and raises SKA scores, with qualitative results indicating better adherence to stroke structure and semantics. The work is promising but constrained by data coverage, evaluation breadth, and reliance on VLM-assisted curation.

**Strengths:**

1. Quality. The technical description is detailed and concrete, including the conditioning mechanism, classifier-free guidance, and the SKA-DPO objective. Training protocols and ablations (e.g., stroke guidance aiding convergence; the DPO margin study) are appropriate. Quantitatively, Stroke3D improves CD metrics and SKA scores over strong baselines.
2. Clarity. The end-to-end pipeline is clearly presented with informative figures (data preparation and overall architecture), and the contributions are explicitly itemized. The appendices document alignment and data-processing choices relevant for reproducibility.
3. Significance. A stroke-and-text interface can lower the barrier to authoring rigged, animation-ready assets. The TextuRig curation and alignment-guided preference optimization are broadly useful ideas for skeleton-to-mesh generation.

**Weaknesses:**

1 Data dependence and coverage. Performance and generalization rely on curated sets (MagicArticulate, SKDream, and TextuRig). The paper acknowledges dataset limitations and sensitivity to rare concepts (e.g., plants before DPO). Scaling and coverage remain open issues.
2 Stroke simulation vs. real inputs. Structural conditioning is trained on perturbed 2D projections of 3D skeletons rather than large-scale human sketches. This domain gap may reduce robustness to messy real drawings; analysis of viewpoint or stroke ambiguity is limited.
3 Evaluation breadth. Skeleton evaluation is primarily CD-based and mesh alignment uses SKA on a 108-sample set following SKDream. There are no user studies or downstream animation stress tests (e.g., auto-skinning stability under motion), and comparisons to recent autoregressive skeleton generators under stroke constraints are sparse.
4 Ablations and controls. While DPO-margin and stroke-conditioning ablations are provided, the incremental effects of TextuRig versus SKA-DPO are not fully disentangled across categories, and variance over random seeds is not reported.

**Questions:**

1. Robustness to real sketches. How does performance change with real, noisy human strokes (varying density, missing joints, occlusions)? Please provide sensitivity curves versus stroke sparsity/noise.
2. Generalization to rare/unseen concepts. Can you quantify success rates for long-tail categories (e.g., “samurai,” “turtle”) and the effect of prompt phrasing beyond qualitative examples?
3. Downstream animation quality. Beyond SKA/CD, do the produced meshes remain stable under standard auto-skinning and motion-retargeting tests (e.g., penetration, joint collapse) across sequences?

---

> ### Author Response · Authors · 2025-11-22
> **Answer to weakness1**
>
> We sincerely thank you for your recognition of the **technical quality, the clarity of our presentation, and the significance of our work**. Thank you very much for your valuable comments!
> We will address all the concerns point by point.
>
> **W1**: Concern of data dependence and coverage.
>
> **Answer to W1:** Thank you for your suggestion. We would like to clarify that our framework's **strong generalization** capability is not just from our curated data, but is fundamentally rooted in our **use of large-scale public datasets and powerful pre-trained foundation models**. And its performance is improved by our **special designed optimization stategy.**
>
> 1. **Reliance on Large-Scale Public Dataset.** As detailed in Section 4.1 in origin version PDF, we employed the MagicArticulate dataset. To our knowledge, this was the **largest and most comprehensive public dataset** of paired skeletons and meshes available at the time of submission. Its large scale provides a strong foundation for validating model generalization. Our experiments confirm that for categories well-represented in these datasets (e.g., humans, animals), our model demonstrates robust generalization.  As explicitly analyzed in Appendix B.3 in origin version PDF, "the primary bottleneck lies in the dataset" rather than the model architecture. We firmly believe that our framework is scalable; given its strong performance on learned categories, it possesses the capacity to cover rare concepts as more comprehensive 3D datasets become available in the future.
>
> 2. **Foundation Model Priors Guarantee Semantic Generalization.** For the mesh synthesis stage, our model is not trained from scratch. As stated in Section 4.1（line372 of origin version）, our model builds upon SKDream, **which is initialized with pre-trained MVDream weights.** MVDream itself is derived from large-scale **pre-trained models (Stable Diffusion 2.1)**, meaning our framework implicitly inherits strong, generalizable priors from massive, diverse datasets.
>
> 3. **Novel optimization strategy for skeleton-mesh alignment.** Regarding the sensitivity to rare concepts, we respectfully view the "plants" category as strong evidence for our SKA-DPO method's efficacy. As shown in Table 2, the baseline model struggled significantly with "Plants" (SKA score: 53.53). This weakness was not resolved by data augmentation alone (+TextuRig score: 53.99). This is precisely why we introduced SKA-DPO. By applying our preference optimization, the "Plants" score surged to 70.63. This demonstrates that **our method is not a victim of this data weakness but is, in fact, an effective solution to address it.**
>
> However, to further enhance the readability and clarity of our manuscript regarding the sources of our generalization capability, we have made specific revisions (**highlighted in blue**). In **Lines 318-319**, we explicitly detail MagicArticulate as the largest and most comprehensive public benchmark available to underscore the scale of our structural priors. Additionally, in **Lines 353-356**, we have clarified that our mesh synthesis module is initialized from powerful foundation models (MVDream/Stable Diffusion), ensuring that our framework inherits strong semantic generalization capabilities beyond the curated training set.

---

> ### Author Response · Authors · 2025-11-22
> **Answer to Weakness2**
>
> **W2:** Concern regarding the domain gap of simulated strokes and the limited analysis of viewpoint.
>
> **Answer to W2:** We thank the reviewer for highlighting these important challenges. **We would like to clarify that our framework possesses robustness to real-world drawings.**  We will address these points in two primary ways: first, through **the design of our canvas tool** which inherently bridges the domain gap, and second, through **qualitative examples and additional experiments in the paper** that show our model handling varied and imperfect user inputs.
>
> 1. **Bridging the Domain Gap via Constrained Tool Design.** As seen in Figure 1 of our original paper, all generation examples (including the human, dinosaur, bird, and tree) were **created by humans using our provided canvas tool**, not by simulated data. Our canvas tool is **motivated by professional 3D software** (like Blender), where skeletons are defined by joints and connected line segments ("bones"). We replicated this structure: our tool guides users to **draw with 2D connected line strokes rather than free-form sketches.** This design choice inherently aligns the structure of "real" user inputs with our graph-based training data, significantly narrowing the perceived domain gap from the outset. Through this tool design, our training process was designed to anticipate real-world imprecision. As described in Section 3.3 of origin version, we simulate stroke data by "apply perturbations to 2D projections, which **mimics the imprecision inherent in hand-drawn strokes**". This step explicitly trains the model to handle the uncertainty of human drawing.
>
> 2. **Empirical Evidence:** We would like to highlight that our original analysis provided evidence for the model's ability to handle stroke ambiguity and varied viewpoints. As discussed in Appendix B.2 and shown in Figure 8(c) of origin PDF, the 'fox' example demonstrates strong generalization. The model successfully generates a logically consistent skeleton even from a vertically flipped stroke, proving it learns a robust prior that generalizes beyond simple projections and can handle significant structural ambiguity. Furthermore, Figure 1 demonstrates practical application on different views, such as a human from a side-view ('A man is running...') and a man from a frontal-view ('A man is standing.'). And the plant and the dinosaur has stroke ambiguity.
>
> However, we agree that explicit clarification and further empirical validation would strengthen the paper. We have added a detailed description of our canvas tool's design in **Lines 362-370 (highlighted in blue)** to clarify how it inherently aligns user inputs with skeletal graph topology . Additionally, we have **revised the caption of Figure 1 (highlighted in blue)** to explicitly state that these results are practical applications generated from real human-drawn strokes. Furthermore, to rigorously demonstrate the model's robustness against input perturbations and varying viewpoints (rotations), we are conducting additional experiments. We present these new qualitative visualizations in **Appendix Figure 11** of the revised manuscript.

---

> ### Author Response · Authors · 2025-11-22
> **Answer to Weakness3**
>
> **W3**: Concern regarding the limited breadth of evaluation, and sparse comparisons to stroke-constrained autoregressive generators.
>
> **Answer for W3:** We thank the reviewer for these suggestions regarding evaluation breadth. We respectfully argue that our current evaluation suite is **rigorous and fully aligned with established community standards.** Specifically, we employ Chamfer Distance (CD) for skeleton accuracy and SKA Scores for mesh alignment—both of which are universally accepted metrics in this research community for benchmarking geometric fidelity and alignment. However, to address the specific concerns regarding downstream animation stability and the landscape of autoregressive baselines, we have conducted **additional stress tests** and provide a detailed clarification below.
> 1. **Robust Auto-Skinning Stability Confirmed via VLM-Based Stress Tests.** Regarding the concern about auto-skinning stability, we first highlight that our **Figure 1** already provides an initial demonstration of an animated rabbit, which was processed using standard automatic skinning tools in Blender without exhibiting penetration artifacts. We respectfully clarify that this stability is a direct result of the high skeleton-mesh alignment achieved by our SKA-DPO strategy. Since automatic skinning is a mature technology in professional software (e.g., Blender), our work intentionally focuses on generating structurally coherent assets that are immediately compatible with these standard tools, rather than optimizing the skinning algorithm itself.
>
>     However, we appreciate the reviewer's excellent suggestion to expand this into a formal "stress test." Since there are no established quantitative metrics for evaluating specific skinning artifacts (e.g., mesh penetration or collapse) in generative tasks, **we designed a novel VLM-based evaluation protocol** to serve as an objective proxy for user assessment: we selected three representative samples each from the **human, animal, and plant categories**. We applied automatic skinning to rig them and then animated them with several motions. We rendered frames from these animations and used Gemini to evaluate the skinning quality, specifically checking for artifacts like unnatural deformation, penetration, or collapse.
>
> 2. **Absence of Existing Stroke-Constrained Autoregressive Baselines.**
> Regarding the comparison to "recent autoregressive skeleton generators under stroke constraints," we respectfully wish to clarify the state of the field. To the best of our knowledge, Stroke3D **is the first framework to generate rigged 3D models using 2D strokes as an explicit structural constraint.** In preparation for this work and this rebuttal, we have conducted a thorough survey of recent literature and **have not found any SOTA autoregressive skeleton generators (such as MagicArticulate or UniRig ) that are designed to accept 2D strokes as input;** they operate on 3D meshes or point clouds. Therefore, a direct comparison "under stroke constraints" was not feasible. **If the reviewer is aware of such methods, we would be grateful for the references.** We would be very interested in evaluating against them, as they would be highly relevant to our work. We would be grateful for the references and would be eager to evaluate against them in our future work, as they would be highly relevant.
>
> We thank the reviewer for pushing us to validate the downstream utility of our assets.  We instructed the VLM to rate each frame on a scale from 1 (severe failure) to 5 (flawless) across two distinct metrics: Non-Penetration and Structural Integrity (No Joint Collapse).
>
> | Category | Avg. Non-Penetration Score (1-5) | Avg. No-Collapse Score (1-5) |
> | :--- | :---: | :---: |
> | Human | 5.00 | 4.96 |
> | Animal | 5.00 | 4.93 |
> | Plant | 4.90 | 4.85 |
>
> It can be seen that our results are robust under animation.
>
> And we have added a qualitative result in **Figure 8**, demonstrating the stability of our models under motion.

---

> ### Author Response · Authors · 2025-11-22
> **Answer to Weakness4**
>
> **W4:** Concern regarding the incomplete disentanglement of ablation components (TextuRig vs. SKA-DPO) and the lack of variance analysis.
>
> **Answer for W4:** We respectfully clarify that **our experimental design already explicitly addresses both the disentanglement of contributions across categories and the management of seed variance through rigorous averaging protocols.** We detail these two aspects below:
>
> 1. **Incremental Effects Are Explicitly Disentangled in Table 2 of origin PDF.** We respectfully point the reviewer to **Table 2**, which was specifically designed to provide the exact disentanglement of TextuRig versus SKA-DPO across all categories. As shown in the table:
>
>     - The "SKDream" row establishes the reference performance.
>     - The "+TextuRig" row isolates the incremental improvement gained solely from SFT with our curated dataset.
>     - The "+SKA-DPO" row isolates the effect of applying only our preference optimization strategy.
>     - The "Ours" row demonstrates the complementary boost when both components are integrated.
>
>     This factorial design clearly quantifies the impact of each component across every reported category (Character, Animal, Plant, etc.), demonstrating that while TextuRig provides a solid foundation, SKA-DPO yields critical improvements in structural alignment.
>
> 2. **Variance Is Mitigated via Standard Multi-Seed Evaluation Protocols.** Regarding variance over random seeds, we strictly adhered to the official evaluation protocol established by the **SKDream** baseline to ensure fair comparison. For every test sample, this protocol requires generating **4 candidates using different random seeds** and reporting the **averaged score**.
>
>     Therefore, the metrics reported in Table 2 are not single-seed snapshots but are already averaged results that account for variance. This ensures that the reported improvements are statistically robust and not artifacts of lucky seeding.
>
> We thank the reviewer for pointing out that these methodological details could be stated more explicitly. We have updated **the caption of Table 2, highlight in blue,** in our revised manuscript to clarify the distinct roles of the ablation components. Furthermore, to ensure transparency regarding our experimental rigor, we have expanded the "Evaluation Metrics" paragraph in **Section 4.2 (Lines 428-433, highlighted in blue)** to explicitly describe the multi-seed averaging protocol used to handle generation variance.

---

> ### Author Response · Authors · 2025-11-22
> **Answer to question1**
>
> **Q1:** Robustness to real sketches. How does performance change with real, noisy human strokes (varying density, missing joints, occlusions)? Please provide sensitivity curves versus stroke sparsity/noise.
>
> **Answer to Q1:** We thank the reviewer for raising this crucial point regarding robustness to real-world input imperfections. We have addressed this through both qualitative demonstration and quantitative stress testing.
>
> 1. **Robustness to Real Human Strokes (Qualitative Analysis):** As clarified in our Response to Weakness 2, our system utilizes a dedicated Canvas Tool designed to bridge the domain gap between freehand sketches and graph structures. We emphasize that all results in Figure 1 of the main paper were generated from real human inputs collected via this tool. To further demonstrate resilience to noisy inputs, we have **added Figure 11 in Appendix D**. This figure explicitly visualizes generations conditioned on:
>
>     - User-drawn strokes with natural jitter.
>
>     - Perturbed inputs (simulating noisy detection).
>
>     - Novel viewpoints, proving that our model does not overfit to canonical poses.
>
>
> 2. **Sensitivity to Sparsity and Missing Joints (Quantitative Curve):** To directly address the request for a sensitivity analysis against stroke sparsity (missing joints/occlusions), we performed a quantitative stress test by randomly dropping joints from the input strokes and measuring the reconstruction error (Chamfer Distance). We have **added Figure 9 (Section 4.4)** to report these results. The sensitivity curve demonstrates that our method maintains robust performance (low CD scores) even as the number of dropped joints increases. The error rises gradually rather than catastrophically, indicating that Stroke3D effectively leverages learned structural priors to hallucinate plausible geometries even when input guidance is sparse or occluded.

---

> ### Author Response · Authors · 2025-11-22
> **Answer to question2**
>
> **Q2**: Generalization to rare/unseen concepts. Can you quantify success rates for long-tail categories (e.g., “samurai,” “turtle”) and the effect of prompt phrasing beyond qualitative examples?
>
> **Answer to Q2:** We appreciate the reviewer's inquiry regarding generalization to long-tail concepts. We have addressed this through both new quantitative metrics and structural stability tests.
>
> 1. **Quantifying Success on Long-Tail Categories:** Dataset-Level Quantification: As detailed in the **new Table 5 (Appendix D)**, we conducted a quantitative evaluation on specific long-tail categories, including "Mythical Creatures", "Toys" and "weapons". Stroke3D consistently outperformed baselines, achieving a significantly lower Chamfer Distance. This statistically proves our generalization capability on unseen topologies.
>
> 2. **Case-Specific Stability:** For the specific examples of "Samurai" and "Turtle" **shown in Figure 11**, we performed a stability test using 10 different random seeds for each concept. We observed a 100% success rate in generating valid, rigged meshes that aligned with the input strokes. Consequently, the geometric generation is highly robust to prompt variations (e.g., switching "Samurai" to "Warrior" or "Human"). The structure remains "locked" by the strokes, preventing the geometric collapse often seen in pure text-to-3D methods when dealing with rare prompts.

---

> ### Author Response · Authors · 2025-11-22
> **Answer to question3**
>
> **Q3:** Downstream animation quality. Beyond SKA/CD, do the produced meshes remain stable under standard auto-skinning and motion-retargeting tests (e.g., penetration, joint collapse) across sequences?
>
> **Answer to Q3:** Like the answer to W3, we condact a VLM-based evaluation that show our mesh is robust under animation, and more cases can be seen in **Figure 8** in the new PDF.

---

### Author Response · Authors · 2025-11-27
**Looking Forward to Your Feedback**

Dear Reviewers,

We are highly grateful for your time and effort to review our work! We understand that you may have busy schedules, but we greatly value your feedback. And we are pleased that reviewers are excited about the novel contributions of our work. Reviewers remark that **“the significance of stroke-and-text interface can lower the barrier to authoring rigged, animation-ready assets.”**[zQuz], **“the paper addresses an interesting and novel task: generating 3D articulated objects from a combination of 2D strokes and text descriptions. ”**[X4bw], **“the system demonstrates both structural control and high visual quality”**[6nvo], **“the paper tackles an interesting and novel task: generating skeleton-mesh pairs from 2D strokes and text prompts, which has not been explored in prior work. ”**[vd39].

We thank the reviewers for the strong praise of our work and contributions. **Our aim is to gain insights into whether our responses effectively address your concerns and to address any additional questions or points you may have.**

We eagerly look forward to the opportunity for further discussion with you. Thank you for your thoughtful consideration.

Best regards,

The Authors

---

### Author Response · Authors · 2025-12-01
**Rebuttal Summary to AC**

Dear Area Chair,

We sincerely thank you for handling our submission, and we are grateful to all reviewers for their valuable time and constructive feedback. Their comments have significantly improved our paper. Below we summarize the key contributions and our responses to all reviewer concerns.

**Summary of Contributions:**

1. **Pioneering a Novel Task.** We are the first to address the challenge of generating 3D articulated objects from 2D strokes and text descriptions. By significantly lowering the barrier to authoring rigged assets, our work holds substantial practical value. **(Novelty and significance acknowledged by all Reviewers zQuz, X4bw, 6nvo, and vd39).**
2. **Fundamental novelty in algorithmic design:** Our work establishes a fundamentally new paradigm for rigged asset generation. We introduce a **novel two-stage framework** to resolve the "unpredictable results" caused by the lack of structural skeleton control in prior methods. Specifically, we are the first to apply a **Latent Graph Diffusion Model (Sk-DiT)** to synthesize 3D skeletons from 2D strokes and text. Following this, we employ an enhanced skeleton-to-mesh method to generate geometry that corresponds accurately to the skeleton.  **(Acknowledged by Reviewers X4bw, vd39).**
3. **Contribution in dataset curation and  optimization strategy for mesh synthesis.** We constructed **TextuRig**, a high-quality dataset that resolves a critical deficiency in the field: the lack of captioned pairings between high-quality skeletons and rich textured mesh. TextuRig supports the robust learning of our mesh synthesis module. **(Acknowledged by Reviewers X4bw, 6nvo).** To further improve our skeleton-to-mesh model, we design the SKA-DPO, which using of the Skeleton-Mesh Alignment Score as the preference signal. This specific strategy directly ensures the generated mesh aligns perfectly with the underlying skeleton constraint. **(Acknowledged by Reviewers zQuz, X4bw).**

**Summary of Revisions**

We set Reviewer zQuz: R1,  X4bw: R2, 6nvo: R3, vd39:R4

All concerns raised by reviewers have been addressed point-by-point, including those related to robustness, data curation, evaluation breadth, and methodological details. We incorporated the following **key updates** into our revised manuscript (highlighted in **blue** in the updated paper):

- **Clarification on Data Curation & Generalization (R1W1, R1W4, R2W1, R2Q3, R4W2, R4W4, R4W5, R4Q4).** We clarified the construction logic and usage of our **TextuRig** dataset (**Lines 95-99, 289-290**), and the independence of our skeleton data sources (**Lines 318-323**). We further detailed that mesh synthesis module is initialized via MVDream (Stable Diffusion 2.1), ensuring strong generalization capabilities (**Lines 353-356**). Multi-seed variance analysis has been explicitly updated to confirm result stability (**Lines 430-433**).
- **Expanded Evaluation on Stability & Alignment (R1W3, R1Q2, R1Q3, R2Q2, R4W3).** We significantly expanded the evaluation suite. We added **Table 6** to quantify the precise alignment between 2D strokes and 3D skeletons, and **Fig. 8 & Fig. 9** to demonstrate animation stability and robustness to joint dropout. Furthermore, we verified performance on **Rare Concepts** to prove the system does not overfit to common categories (**Appendix Table 5 and Fig. 11**).
- **Enhanced Robustness to Real-World Inputs (R1W2, R1Q1, R2Q1, R3Q2).** We validated our model against real-world user strokes and varying viewpoints to address concerns on generalization. New experiments and detailed descriptions of our canvas tool confirm that the model handles diverse drawing styles and camera angles effectively, demonstrating strong robustness in practical scenarios (**See Fig. 1 Caption, Lines 362-370, and Appendix Fig. 11**).
- **Clarification on Algorithmic Design & Terminology (R3W2, R4W1).** We explicitly clarified the pioneering nature (**Lines 74-75**) of our Sk-VAE design, inspired by T2I Diffusion Models (**Line 245**). We also integrated standard automatic skinning into our definition of "Rigged Model" to address terminology concerns (**Lines 105-107**). Additionally, we revised the text to strictly state that text prompts serve as mandatory active conditions for mesh synthesis (**Lines 289-290**).
- **Clarification on Baselines & Data Processing (R3W1, R3Q1).** We provided an explicit rationale for the selected baselines in **Lines 380-382** to ensure fair evaluation. We also clarified details regarding the projection method (orthographic transformation) in **Lines 455-456**. **Appendix Fig. 11** provides visual evidence of our model's robustness to varying camera viewpoints.
- **Citations & Presentation Improvements (R4Q1, R4Q2, R4Q3).** We incorporated missing citations (**Lines 185-186**) and improved the layout and grammar in the revised manuscript to enhance readability.

We believe we have thoroughly addressed all reviewer concerns and hope the AC finds our responses satisfactory.

---

### Meta-Review · Area_Chair_KC7x · 2026-01-06

**Summary:**

The paper presents a method that produces rigged 3D mesh from user-drawn 2D strokes and a text prompt. It builds upon a prior work SK-Dream (CVPR-2025), which generate rigged 3D mesh from 3D stroke and text prompt. They extend with a 2D stroke conditioned 3D-stroke DiT generation model, and also introduces a novel geometric-based DPO fine-tune stage. The paper initially received two rating 6 and one rating 4 and one rating 2. The reviewers acknowledged that 1) the proposed task is novel and no one has tried to address before; 2) delivers effective results for skeleton-mesh generation. The main critical concerns are 1) the limited technical novelty; 2) lack of additional evaluations, such as 2D strokes from human users etc.; 3) data dependence and coverage. The authors provide detailed clarifications regarding the raised concerns. Although parts of it improve upon SKDream, the integration of 2D stroke control and the novel DPO alignment strategy delivers impressive performance for this new task. After carefully weighing the paper's strengths against its limitations, the AC recommends acceptance of the paper.

**Reviewer Concerns:**

The AC think most of the concerns have been clarified.

**Reviewer Scores:**

The AC think the reviewers who initially gave negative scores, might upgrade their scores if further discussions are conducted.

---

### Decision · Program_Chairs · 2026-01-26

Accept (Poster)